# Position: Interestingness is an Inductive Heuristic for Future Compression Progress

**Vincent Herrmann** [1 2]   **Jürgen Schmidhuber** [1 2]

## Abstract

This position paper argues one of the bottlenecks on the way towards recursively self-improving systems is the challenge of *interestingness*: the ability to prospectively identify which tasks or data hold the potential for future progress. We formalize interestingness as an inductive heuristic for future compression progress and investigate its predictability using tools from Kolmogorov Complexity and Algorithmic Statistics. By analyzing complexity-runtime profiles under Length, Algorithmic, and Speed priors, we demonstrate that the *inductive property of interestingness*—the capacity for past progress to signal future discovery—is theoretically viable and empirically supported. We prove that expected future progress depends exponentially on the recency of the last observed breakthrough. Furthermore, we show that the Algorithmic Prior is significantly more optimistic than the Length Prior, yielding a quadratic increase in expected discovery for the same observed profile. These findings are experimentally validated across three diverse universal computational paradigms.

## 1. Introduction

Put concisely, this work argues that **interestingness is an essential ingredient for open-ended learning, recursive self-improvement, and artificial general intelligence. It can be described as a heuristic assessing the potential of future compression progress.** We give reasons for assuming that it possesses inductive properties, meaning past progress can be a predictor of future insight. There is important work to be done on finding practical interestingness

[1]The Swiss AI Lab IDSIA/USI/SUPSI, Lugano, Switzerland [2]King Abdullah University of Science and Technology, Thuwal, Saudi Arabia. Correspondence to: Vincent Herrmann <vincent.herrmann@idsia.ch>.

*Proceedings of the 43rd International Conference on Machine Learning*, Seoul, South Korea. PMLR 306, 2026. Copyright 2026 by the author(s).

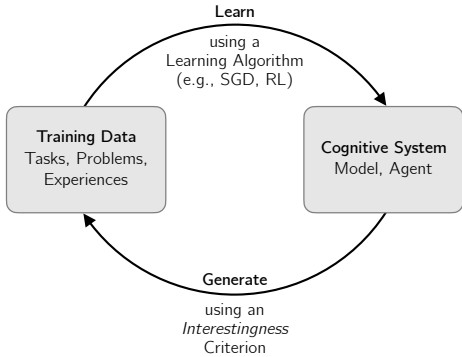

*Figure 1.* Minimalist depiction of a self-sustaining learning cycle. The Learning phase derives new skills or patterns from data, while the Generation phase creates novel artifacts. To sustain true open-endedness, the generation process must be guided by a criterion that distinguishes learnable structure from noise or already-acquired knowledge.

measures, and investigating the exact algorithmic properties of objects and environments that allow continued learnability.

A common notion of how a general recursively self-improving open-ended intelligence might be achieved is, in very simple terms, as a cycle between two alternating phases (see Figure 1): In the learning phase, a system uses a learning algorithm—e.g., gradient descent or Reinforcement Learning (RL) techniques—to extract patterns and regularities in the available data, or to gain the skills necessary to solve available problems and tasks. In the generation phase, the insights from the learning phase are used to produce novel artifacts (synthetic data, new tasks, hitherto unsolved problems). From these, the system can then again learn new capabilities, ideally leading to never-ending progress. To achieve true open-endedness, this cycle must be fully autonomous. It cannot rely on human-in-the-loop filtering, hand-crafted curricula, or synthetic data created by researchers. One example of such a system would be a Large Language Model (LLM) that is iteratively trained on self-generated data—see, for example Wang et al. (2023); Zelikman et al. (2024). It is well known that a naive real-

ization of such a setup leads to model collapse (Shumailov et al., 2024; Dohmatob et al., 2024). However, there exist efforts to modify the generation process, or filter the generated data in such a way that continued learning is possible (Lin et al., 2024; Herrmann et al., 2025; 2026). Another example would be an RL agent that pursues its own goals in an environment. The goals are chosen according to some intrinsic motivation criterion (Schmidhuber, 1991c; Pathak et al., 2017; Colas et al., 2022).

With the success of large-scale models trained with gradient descent and RL techniques, it can be argued that we have a good handle on the learning phase. Developing a generation phase that leads to continued progress, on the other hand, remains an open problem. In the framing of Hughes et al. (2024), how can we generate artifacts that are both *novel and learnable*? What criterion allows us to select the samples, problems, tasks from which the system can learn something meaningful? How can we distinguish between what is unlearnable, already learned, and so-far unknown but learnable? How can we tell if such an artifact is *interesting*?

In this paper, we argue that interestingness is a prospective inductive heuristic for future compression progress. Previous measures of interestingness often provide a post-hoc account of what was learned, while a self-improving system requires a way to predict what will be learned *before* committing significant computational resources. Our contributions are as follows: **(1)** We formalize the Inductive Property of Interestingness using Complexity and Runtime profiles, showing that future compression breakthroughs can be statistically predicted from the trajectory of past progress. **(2)** We prove that the primary predictor of future progress is stagnation length: the probability of a new compression "drop" vanishes exponentially as the gap since the last progress increases. **(3)** We analyze how different priors over computable objects—Length, Algorithmic, and Speed—govern this predictability. We demonstrate that the Algorithmic Prior is significantly more "optimistic" than the Length Prior, yielding a quadratic increase in expected future progress for the same observed profile. **(4)** We provide empirical validation across three computationally universal paradigms (2-Tag systems, Rule 110 cellular automata, and Brainfuck), confirming that these theoretical trends hold in physically realizable regimes.

**Alternative View: *Open-endedness emerges from social multi-agent interactions.***
A compelling alternative is that Interestingness is not treated as an intrinsic property of objects, or a relational property between a subject and an object, but an emergent phenomenon of social interaction. Such settings span from competitive two-player games (Schmidhuber, 1997; 1999) to complex multi-agent ecosystems involving theory-of-mind and embedded agency (Meulemans et al., 2025). There, learning potential is often quantified through disagreement: if a population of models exhibits high variance in their predictions of an object, that object is deemed to contain unresolved, learnable structure (Pathak et al., 2019; Shyam et al., 2019; Sekar et al., 2020; Sancaktar et al., 2022). Similarly, the notion of impressiveness (Lehman & Stanley, 2012) suggests that an artifact is interesting if it serves as a "proof of work"—a hard-to-reach state that signals computational effort to other agents. While we recognize the power of these social dynamics, we argue they do not bypass the fundamental problem of interestingness, they merely decentralize it. Whether an agent is filtering synthetic data or evaluating a peer's "impressive" achievement, it is still performing a prospective assessment of future progress. Even a co-evolutionary ecosystem (Pugh et al., 2016; Brant & Stanley, 2017) can be viewed as a single macro-agent in a self-generating loop. Therefore, we believe that the inductive properties of learnability we analyze in the following sections remain relevant to these social systems.

## 2. Interestingness as Expected Compression Progress

We refer to the artifacts a cognitive system (the *subject*, or model) learns from as *objects*. This highlights the fact that interestingness is not necessarily an intrinsic property of an object in isolation, but rather a relational property between the object's structure and the subject's current internal state. Prior literature has approached interestingness from various angles. An object can be deemed interesting if it is unfamiliar, having not been encountered as often as other objects (count-based, space coverage), if it is surprising, not correctly predicted by a model (prediction error, adversarial), if it changes a model's predictions or beliefs (information gain), if it improves performance along some metric (learning progress), or if it allows a model to further compress a history of data (compression progress). We refer the reader to Appendix A for a more detailed review.

Some of these criteria—namely, information gain, learning progress, and compression progress—are closely related: they all measure shifts in the subject's uncertainty or description length. Information gain provides a probabilistic view of these shifts, learning progress provides an optimization-based view, and compression progress provides an algorithmic view. They provide an *introspective* account, subjective to the system, of how much insight has been gained. There is significant overlap, but no notion completely subsumes any other: Not all information gain is necessarily compression

progress, and information gain (as opposed to compression progress) does not usually take the description length of the model itself into account. Learning progress can be defined for various loss functions—not just NLL—and also ignores model size. For the theoretical analysis in this work, we choose compression progress, since it allows us to use the tools of Algorithmic Information Theory (AIT). Despite their subtle differences, we expect that our results for compression progress largely transfer to other notions of interestingness.

An important drawback of all these criteria is that they assume that the computational effort has already been spent. In an open-ended setting, this assumption is untenable: the generation phase must decide where to spend effort before the outcome is known. This means a useful interestingness criterion must be *prospective*. It is not enough to know how much has been learned; we need to know how much we can still learn. The criteria above are largely post-hoc: they quantify the progress made after the computational effort of training has been expended. This makes them insufficient for the "generation phase," where the system must select promising objects from a vast space of possibilities before committing significant resources to them. In everyday speech, we often call a thing interesting because we have a feeling that it holds further secrets, because we think that there is more to learn about it. This is the sense of "interesting" we advocate for: *a prospective heuristic that predicts future compression progress, based on past experience.* This leads to a fundamental question: to what extent and under which conditions is such a prediction even possible? The following sections provide a formal treatment of this inquiry, investigating the algorithmic conditions under which future progress can be rigorously predicted.

**Can We Learn How to Predict Learnability?**   Before moving to the formal investigation, we consider whether the prediction of compressibility can be treated as just another learning problem. As described, information gain, learning progress, and compression progress measures require running the inference or learning algorithm to evaluate an object. This makes them not directly usable in an open-ended setting: it is not possible to simply "train on everything" to see what works. Doing so is not only computationally prohibitive but also risks model collapse, as mentioned before. Instead, we want to *predict* these measures before investing the effort to learn from them. The natural approach when trying to predict "post-hoc" data is *learning* how to predict it. This is indeed what happens when compression progress is used as an intrinsic reward for an RL agent, such as in a Controller-Model setup (Schmidhuber, 2015; Kompella et al., 2017). Optimal Bayesian exploration in dynamic environments (Sun et al., 2011) formalizes the expected information gain as a "curiosity" Q-value. However, the cost

of computing an optimal policy is exponential in the number of look-ahead steps, making it infeasible in many realistic settings—again, requiring a learnable approximation.

Meta-cognitive properties like interestingness and curiosity values can be difficult to learn for several reasons: **Non-stationarity**—Interestingness is a moving target. As the subject learns, an object that was once insightful becomes boring, requiring the meta-model to constantly adapt to the subject's changing state. **Data Sparsity and Cost**—Ground-truth labels for progress are expensive to obtain, as they might require executing a full training phase just to evaluate a single data point or task. **Credit Assignment**—In the many batches of data required for training, it is profoundly difficult to identify which specific samples contributed to the reduction in loss. This credit assignment problem makes it hard for a predictor to associate specific object features with the resulting learning progress.

> **Alternative View:** *We can predict human-distilled Interestingness via LLMs.*
>
> A compelling alternative to explicit algorithmic measures is the use of LLMs as a "Model of Interestingness". The OMNI framework (Zhang et al., 2023; Faldor et al., 2025) argues that because LLMs are trained on large amounts of human-generated data, they have already internalized nuanced human notions of what is worthwhile and novel. This has been shown to work in practice. But how far do we expect this approach to generalize? OMNI fundamentally emulates human capacity for nuanced judgment; it effectively asks: "What would a human want to learn next?" But what yields learning progress for a human is not necessarily identical to what yields progress for an artificial subject. Humans and our current AI systems possess vastly different learning architectures and existing knowledge bases. Perhaps we will eventually converge to a shared conception of interestingness among general intelligences—whether artificial or human. But if the goal is to *reach* general intelligence through an open-ended process, we cannot assume that the current state of LLMs is already sufficiently close to possess such a universal notion of interestingness. Furthermore, LLMs currently have no privileged insight into their own internal compression frontiers—they can assess data solely based on an external heuristic rather than an introspective assessment of their own potential for insight. True autonomous open-endedness likely requires the ability to identify potential progress in domains that human intuition has not yet charted. There it is not clear, to what extent an LLM can distinguish potentially promising avenues from dead ends.

We have made informal arguments why it is difficult to predict future learnability. Essentially, we are asking about

the *inductive property of interestingness*: can we infer how much there is yet to learn, and how easily accessible that learnable content is, based on the trajectory of how much we have already learned? Despite these challenges, the algorithmic structure of the data provides a stable ground—hence our turn to Algorithmic Information Theory.

## 3. Runtime and Complexity Profiles

To analyze interestingness and its inductivity with sufficient generality, we move to an abstract setting using the tools of Algorithmic Information Theory. This section introduces the necessary concepts and results, more details can be found in (Li & Vitányi, 1990) and (Vereshchagin & Shen, 2016). We consider all data and artifacts as binary strings $x \in \{0,1\}^*$. Let $K(x)$ denote the standard prefix-free Kolmogorov complexity of $x$: the length of the shortest program $p$ on a universal prefix Turing machine that halts and outputs $x$. We are not merely interested in the absolute compression limit $K(x)$, but in the compression progress made as a function of computational effort. Let $K^r(x)$ be the time-bounded Kolmogorov complexity: the length of the shortest program that computes $x$ within $r$ steps. This bounded complexity allows us to formalize learning as a trajectory of algorithmic discovery, where increasing computational resources uncover progressively shorter descriptions of the data.

Our system has a history of experiences, already partially compressed into some form $B$. This representation $B$ is the current state of our model. Now the systems observes a new object $o$. A new object is introduced by concatenating it with $B$. By this concatenation, we are asking "is $o$ interesting in the current context?", instead of "is $o$ interesting, in and of itself?". This is exactly the relational property between model and object we mention at beginning of Section 2. Now three qualitatively different things can happen. First, $o$ could be algorithmically random relative to $B$—it carries no learnable regularities, and the compression of $Bo$ behaves as if $B$ alone were being further compressed, with $o$ contributing only irreducible noise. Second, $o$ could be itself compressible—it contains internal regularities that a short program can exploit, yielding immediate but local compression progress. Third, and most interestingly, $o$ and $B$ could be *relationally* compressible: $o$ contains structure that unlocks new descriptions of $B$ itself, or vice versa—$B$ provides context that makes $o$ dramatically shorter to describe. This third case is precisely what we mean by $o$ being interesting in context. It could of course also be some mixture of these three cases. The point is, the interestingness of object $o$, in this framing, is measured by the expected compression progress of $Bo$. For our theoretical analysis, we simply define $x := Bo$. Since we want a *prospective* measure—one that from past compression predicts future

compressibility—we study not only the final compression $K(Bo)$, but the whole trajectory $r \mapsto K^r(Bo)$: how description length decreases as more computational effort is allowed. A drop in this profile at runtime $r$ represents a compression breakthrough achievable with effort $r$. Interestingness is then the question of whether this profile, observed up to a current runtime budget $R$, will continue to drop for larger $r > R$.

**Complexity vs. Runtime Profiles**  To study this discovery process, we use the complexity vs. runtime profile (Vereshchagin & Shen, 2016; Bauwens, 2010; Antunes et al., 2017):

$$D_x := \{(r,c) \mid K^r(x) \le c\}.$$

The boundary of this profile, $\min\{c \mid (r,c) \in D_x\} = K^r(x)$, tracks the shortest description of $x$ available given a runtime budget $r$. An object is *logically deep* (Bennett, 1988) if this boundary continues to drop significantly even for very large $r$. Let $n_x$ denote the length of string $x$, and $k_x$ denote the complexity $K(x)$. The fastest way to compute $x$ is the trivial program "return $x$", which has length $n_x$ plus a negligible constant. Since $k_x$ is the complexity of the shortest program computing $x$, $k_x$ is the complexity value that the boundary of $D_x$ will eventually reach, given sufficient runtime (see Figure 2, top). Our central question is about the inductive properties of the profile: Does the shape of $D_x$ for $r \le R$ for some runtime cutoff $R$ allow us to predict the existence of further "drops" in complexity for $r > R$?

### 3.1. Log-Size vs. Complexity Profiles and Sophistication

To leverage results from algorithmic statistics, we also make use of a related profile: the log-size vs. complexity profile $P_x$:

$$P_x := \{(i,j) \mid \exists A \text{ s.t. } x \in A, K(A) \le i, \log \#A \le j\},$$

where each $A \subseteq \{0,1\}^*$ is a finite set of binary strings. The boundary of $P_x$ represents, for a given set complexity, the log-size of the smallest set containing $x$. From any set containing $x$, it is possible to construct a two-part description of $x$, consisting of the set itself (the model) and an identifier of $x$ within the set (the data-to-model code, also called residual). One simple set containing $x$ is always the set consisting of all strings with length $n_x$. The smallest set is the singleton set containing only $x$ itself, which means it has complexity $k_x$. The boundary of the $P_x$ profile always has a negative slope of at most $-1$, reflecting the trivial exchange between halving the set size at the cost of adding one bit to the description length of the set. A "drop" in $P_x$ occurs when the boundary falls below the line of slope $-1$. Such drops indicate that the corresponding model contains

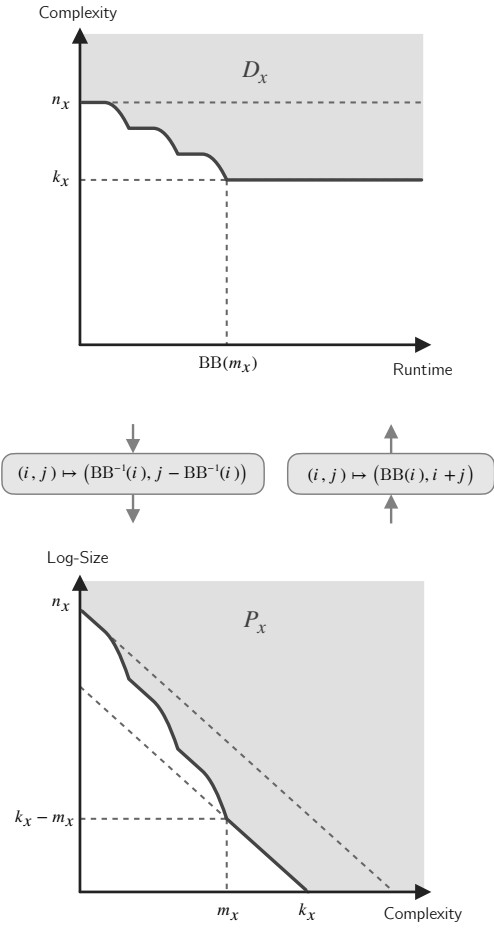

*Figure 2.* Profiles and their relationship. (Top) The complexity vs. runtime profile $D_x$ of a string $x$, starting with minimum complexity $n_x$ for fast programs and dropping to $k_x$ with sufficient runtime. The fastest runtime of a program with length $k_x$ that computes $x$ is $\mathrm{BB}(m_x)$, where BB is the busy-beaver function. (Bottom) The log-size vs. complexity profile $P_x$ of the same string. Both profiles are—up to logarithmic precision—affine transforms of each other, when BB is used to convert complexity values into runtimes.

additional structural information about the string, beyond the random information of the index. The point where the boundary last meets the $-1$ slope line before following it indefinitely defines the *sophistication* of $x$, representing the complexity of all the structural, non-random information contained in $x$ (Koppel, 1987; Antunes & Fortnow, 2009). We refer to the complexity value of this point as $m_x$ (see Figure 2, bottom).

The relationship between complexity vs. runtime and log-size vs. complexity profiles is bridged by the Busy Beaver function $\mathrm{BB}(k)$—the maximum number of steps a halting $k$-bit program can run. BB is uncomputable and extremely fast-growing. But it yields a universal time-scale that abstracts away machine dependence. With this re-scaling, $P_x$

and $D_x$ are approximately affine transforms of each other. The transform

$$(i, j) \mapsto (\mathrm{BB}(i), i + j) \tag{1}$$

converts a log-size vs. complexity profile $P_x$ into a complexity vs. runtime profile $D_x$, within logarithmic precision $O(\log |x|)$ (see Vereshchagin & Shen (2016), Theorems 4 and 6). The inverse transform, from $D_x$ to $P_x$, is of course possible as well. This correspondence is profound: it implies that every drop in $P_x$ (a new non-trivial model with a higher complexity) corresponds to a drop in $D_x$ (a progress in compression when the maximum runtime is increased). Figure 2 illustrates the connection between the two profiles. Many modern machine learning paradigms, including gradient descent optimization, chain-of-thought reasoning, and model scaling, map naturally onto $P_x$ or $D_x$ profiles (see Appendix B for an extended discussion). While real-world algorithms do not precisely follow these theoretical compression limits, these foundational boundaries offer universal insights into the nature of learnable structure.

### 3.2. Counting Strings with Specific Profiles

Results from Algorithmic Statistics quantify how many strings exist that follow a specific profile shape. Let $P$ be a valid log-size vs. complexity profile: an upward-closed set of integer pairs whose boundary has a slope of at most $-1$, or formally, for which $(a, b + c) \in P \implies (a + b, c) \in P \; \forall a, b, c$. For $P$, we define three characteristic values: The log-size value of the leftmost boundary point $n_P := \min\{r \mid (0, r) \in P\}$, the complexity value of the rightmost boundary point $k_P := \min\{r \mid (r, 0) \in P\}$, and the complexity value of the point where the boundary meets the diagonal line leading to $(k_P, 0)$, namely $m_P := \min\{(r, k_P - r) \in P\}$. These values correspond to the length $n_x$, complexity $k_x$, and sophistication $m_x$ of a string $x$ that has a profile $P_x$ of shape $P$.

Vereshchagin & Shen (2016) (Theorem 19) lower bound the number of strings that have a profile close to $P$:

$$\#\{x \mid P_x \approx P\} \geq 2^{k_P - m_P + O(1)}, \tag{2}$$

where the closeness is of order $O(C(P) + \log n_P)$. Theorem 20 of the same work proves the maximum number of strings following a profile close to $P$: There are at most

$$\#\{x \mid P_x \approx P\} \leq 2^{k_P - m_P(\epsilon) + 2\epsilon + O(\log n_P)} \tag{3}$$

strings that are $\epsilon$ close to profile $P$. The value of $m_P(\epsilon)$ is similar to $m_P$.

The first important observation is that for any given valid profile $P$, there exist strings which have a profile close to it. The term $k_P - m_P$ represents the "random" part of a string following profile shape $P$—the bits that cannot be

further compressed into a structural model. Crucially, the number of strings with a given profile is dominated by this residual randomness. This suggests that while structure is rare, strings sharing same structural profile are plentiful, differing only in their structureless noise. The value $n_P$, i.e. the log-length associated with profile $P$, contributes only logarithmically, compared to $k_P$ and $m_P$, to the upper bound of the number of strings. The exact shape of the profile—beyond the values $n_P, k_P$ and $m_P$—only plays a role via its complexity $C(P)$ in the closeness to the lower bound of strings.

## 4. Quantifying the Inductivity of Interestingness

We have defined interestingness as the expected future compression progress when more effort is expended, and are interested in its inductive properties—namely to what extent past progress implies future progress. The concepts introduced in the last section give us a way to quantify exactly this. The proofs here are for log-size vs. complexity profiles, but the $P \leftrightarrow D$ correspondence (Equation 1) allows straightforward conversion of all results to complexity vs. runtime settings.

We formalize the inductivity of interestingness as follows: If we observe a *partial profile* $\hat{P}$ up to (but not including) a complexity cutoff $t$, what can we infer about its continuation? Given the $\hat{P}$, how likely is it that a string $x$ with this partial profile is further compressible? How much compression progress can we expect? When will the next compression progress (i.e., drop in the profile) happen? What is the expected complexity at the last drop ($m_x$) and the expected singleton complexity ($k_x$)? To formulate these expectations, we must assume a prior probability distribution over all possible strings that we might encounter. This prior represents the "world" our learning system inhabits. Since there are infinitely many such computable strings, it is not possible to assign a uniform non-zero probability to all of them. Instead we investigate three very general priors over strings: the Length Prior, the Algorithmic Prior and the Speed Prior. As we will see, the inductivity properties of interestingness depend on the choice of prior.

Analogously to the values $n_P$, $k_P$, and $m_P$ for a full profile $P$, we define the values $\hat{n}$, $\hat{k}$, and $\hat{m}$ for the partial profile $\hat{P}$. That means $\hat{n}$ is the minimal log size at minimal complexity, and the last observed drop lands at the point $(\hat{m}, \hat{k} - \hat{m})$. By saying that we observe, or condition on, a partial profile $\hat{P}$, we mean that the full profile $P_x$ of a string $x$, sampled from a prior distribution, coincides precisely with $\hat{P}$, up to complexity $t$. This means $n_x = \hat{n}$. If there is no further drop at or after $t$, this also means that $m_x < t$, $m_x = \hat{m}$, and $k_x = \hat{k}$. If there is a further drop, then $m_x \geq t > \hat{m}$ and $k_x < \hat{k}$. Figure 3 illustrates this setup. As it turns out,

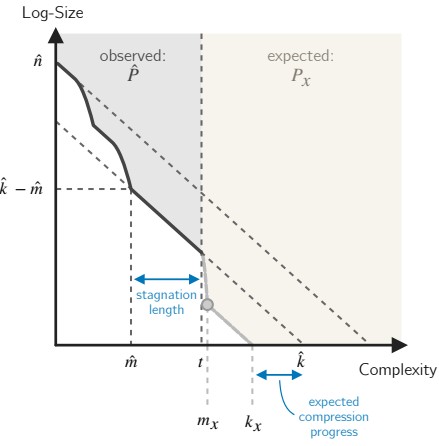

*Figure 3.* The partial profile $\hat{P}$ is defined up to observation cutoff $t$. It has the characteristic values $\hat{n}$, $\hat{k}$, and $\hat{m}$. For a string $x$ with the initial partial profile $\hat{P}$, the expected continuation $P_x$ has its last drop at complexity $m_x$, and a singleton complexity of $k_x$. Theory and experiments show that the stagnation length $t - \hat{m}$ determines the expected compression progress $\hat{k} - k_x$.

the property which—at least for Length and Algorithmic Priors—overwhelmingly predicts the probability of a future drop (i.e., $p(m_x \geq t)$) and of future compression progress (i.e., $\hat{k} - k_x$) is $t - \hat{m}$: the gap between the last observed complexity drop and the observation cutoff. The larger this *stagnation length*, the less likely further compression progress becomes[1]. The bounds from Equations 2 and 3 tell us how many strings exist for a given profile shape. Given a priori probabilities of strings, this allows the calculation of expectations, which we will now do for the three mentioned prior distribution.

### 4.1. Length Prior

The simplest prior considers only string length, effectively asking for the probability that a monkey randomly typing bits generates $x$. To ensure this is a valid semi-measure, we use a prefix-free encoding (e.g., duplicating bits and ending with `01`). With that, the length prior is defined as

$$L(x) := 2^{-(2|x|+2)}. \tag{4}$$

**Proposition 4.1.** *Let $\hat{P}$ be an observed partial profile up to (but not including) complexity $t$, with the last observed drop at $(\hat{m}, \hat{k} - \hat{m})$. Assuming the object $x$ producing $\hat{P}$ is sampled from the Length Prior $L$, the following properties hold:*

*(i) Given $\hat{P}$, the probability of a further drop in the profile*

---

[1]The relationship between $P_x$ and $D_x$ profiles, stagnation length and expected continuation can be explored in the interactive diagrams at https://inductive-interestingness-2026.github.io/future-compression-progress/#theory

$P_x$ at any complexity $\geq t$ is:

$$p_L(m_x \geq t \mid \hat{P}) = 2^{-(t-\hat{m})+O(\log \hat{n})}.$$

*(ii) Given the existence of such a drop and assuming $\hat{k} - t \gg 0$, the expected complexity of $x$ is:*

$$\mathbb{E}_{x \sim L}[k_x \mid \hat{P}, m \geq t] \approx \hat{k} - 2.$$

*(iii) Without conditioning on a further drop, the expected further compression progress is:*

$$\hat{k} - \mathbb{E}_{x \sim L}[k_x \mid \hat{P}] = 2^{-(t-\hat{m})+O(\log \hat{n})}.$$

The proof can be found in Appendix C.1. We can see that the stagnation length $t - \hat{m}$ determines both the likelihood of future compression and the magnitude of expected progress.

## 4.2. Algorithmic Prior

We can explain the Algorithmic or Solomonoff prior (Solomonoff, 1964) in similar terms as the length prior. But instead of the monkey typing the string directly, we now ask for the probability of the monkey typing a program that, when run on a prefix Turing machine $U$, outputs $x$. The Algorithmic prior is

$$M(x) := \sum_{p:U(p)=x} 2^{-|p|}. \tag{5}$$

This prior prefers algorithmically simple strings (ones with a short description length) as opposed to the Length Prior, which prefers short strings.

**Proposition 4.2.** *Assuming the object $x$ producing $\hat{P}$ is sampled from the Algorithmic Prior $M$ rather than the Length Prior, the following properties hold:*

*(i) The probability of a further drop at or after complexity $t$ is:*

$$p_M(m_x \geq t \mid \hat{P}) = \frac{1}{1 + \frac{2^{-\hat{m}+O(\log \hat{n})}}{2^{-t+O(\log \hat{n})}+2^{-\hat{k}+O(\log \hat{n})}}},$$

*which, for $\hat{k} - t \gg 0$, simplifies to:*

$$p_M(m_x \geq t \mid \hat{P}) \approx 2^{-(t-\hat{m})+O(\log \hat{n})}.$$

*(ii) Given the existence of such a drop, the expected complexity of $x$ is:*

$$\mathbb{E}_{x \sim M}[k_x \mid \hat{P}, m \geq t] \approx \frac{\hat{k} + t}{2}.$$

*(iii) Without conditioning on the existence of a further drop, the expected further compression progress is:*

$$\hat{k} - \mathbb{E}_{x \sim M}[k_x \mid \hat{P}] \approx 2^{-(t-\hat{m})+O(\log \hat{n})}.$$

For the proof, please see Appendix C.2. For the Algorithmic Prior, further compressibility is equally determined exponentially by $t - \hat{m}$. However, we can show that it yields higher expected compression than the Length Prior.

**Proposition 4.3.** *Assuming $\hat{k} - t \gg 0$ and $t - \hat{m} \gg \log \hat{n}$, the Algorithmic Prior $M$ yields significantly higher expectations for future insight than the Length Prior $L$. Specifically, the following ratios hold:*

*(i) The ratio between the further drop probabilities is approximately linear in the remaining complexity gap $\hat{k} - t$:*

$$\frac{p_M(m_x \geq t \mid \hat{P})}{p_L(m_x \geq t \mid \hat{P})} \approx \hat{k} - t - 1.$$

*(ii) The ratio between the expected further compression progress is approximately quadratic in the remaining complexity gap:*

$$\frac{\hat{k} - \mathbb{E}_{x \sim M}[k_x \mid \hat{P}]}{\hat{k} - \mathbb{E}_{x \sim L}[k_x \mid \hat{P}]} \approx \frac{1}{4} \left( \hat{k} - t - 1 \right) \left( \hat{k} - t \right).$$

The proof can be found in Appendix C.3. These ratios demonstrate that the inductive property of interestingness—the capacity for past compression progress to signal future discovery—is significantly more robust under the assumption of the Algorithmic Prior. While our primary analysis considers the indefinite future, we show in Appendix C.1.5 and C.2.5 that the probability of discovery is heavily concentrated near the current observation threshold $t$. For any finite computational window, the likelihood of a "breakthrough" vanishes exponentially as the gap since the last progress increases, providing a justification for prioritizing tasks with the most recent "aha!" moments.

## 4.3. Speed Prior

The algorithmic prior $M$ is uncomputable, and it is impossible to obtain exact samples from it. This stems from the fact that the programs generated by a random source may never halt—a property that is generally undecidable. The Speed Prior $S(x)$ (Schmidhuber, 2002b) addresses this by penalizing both program length and runtime. Intuitively, for the Speed Prior, the monkey has to first spell out the maximum runtime in a prefix-free format, and then the program computing $x$. If the program has not halted before the specified runtime, the output is set to the empty string. We write $p \rightarrow_i x$ iff it takes $2^{i-|p|}$ steps for program $p$ to compute output $x$. The speed prior is defined as

$$S(x) := \sum_{i=1}^{\infty} \sum_{p \rightarrow_i x} 2^{-(i+|p|)}. \tag{6}$$

It is closely related to the Levin Complexity $Kt(x) := \min\{|p| + \log \text{time}(p) : U(p) = x\}$ (Levin, 1984).

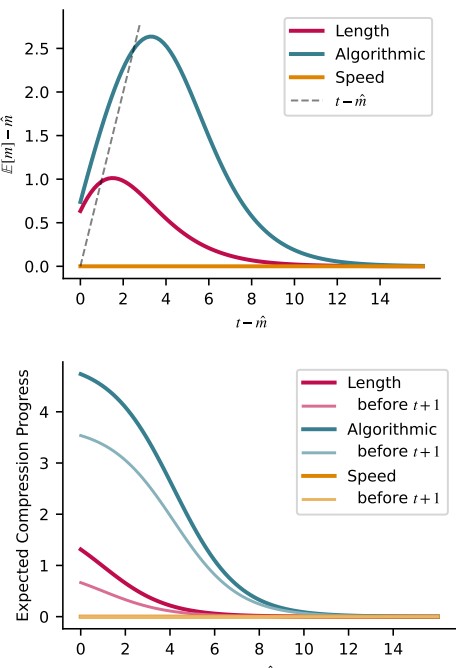

*Figure 4.* The expected difference between the complexity of the last drop $m$ and the last observed drop $\hat{m}$ (top), and the expected future compression progress (bottom). Both are plotted as a function of the stagnation length $t - \hat{m}$, for the Length, Algorithmic, and Speed Prior. The bottom plot also shows the expected progress immediately after the observation cutoff $t$. As we can see, this is where the majority of the expected progress should happen.

**Proposition 4.4.** *Assuming the object producing $\hat{P}$ is sampled from the Speed Prior $S$, the following properties hold:*

(i) *The probability of a further drop at or after complexity $t$ is:*
$$p_S(m_x \geq t \mid \hat{P}) \approx 0.$$

(ii) *Given the existence of such a drop, the expected final complexity is:*
$$\mathbb{E}_{x \sim S}[k_x \mid \hat{P}, m \geq t] \approx \frac{\hat{k} + t - 1}{2}.$$

(iii) *Without conditioning on the existence of a further drop, the expected further compression progress is:*
$$\hat{k} - \mathbb{E}_{x \sim S}[k_x \mid \hat{P}] \approx 0.$$

The proof can be found in Appendix C.4. The Speed Prior is the most "conservative". Because it penalizes runtime, it assumes that if a faster way to compress the string existed, it would have been found already. This prior essentially treats the current frontier of compression as the final one. It should be mentioned that this, at least to a certain extent, is an artifact of the extreme runtimes of the BB function. In practice, more realistic runtimes lead to a milder decrease of expected compression progress.

### 4.4. Discussion: The Inductive Property of Interestingness

With Equation 1, it is possible to transform the above results from $P_x$ profiles to complexity vs. runtime profiles $D_x$, since the strings—and hence also the number of strings—with a certain $P$-profile shape are the same the ones with the corresponding $D$-profile shape. Thus, we see that under standard priors, the "Inductive Property of Interestingness" holds a specific form: past compression progress can be an indicator of future progress, but only if that progress is recent relative to the computational effort expended. As shown in Figure 4, if the stagnation length $t - \hat{m}$ is small, there is an expectation for future drops, at least for the Length and Algorithmic Prior. This may justify investing resources into objects that have recently yielded insight. Conversely, after a long stagnation, the probability of a future "aha!" moment vanishes exponentially. Note, however, that this does not mean that it is impossible: as mentioned in Section 3.2, all valid profile shapes are closely followed by some strings. They are just rare under the priors we investigated. But in general, all else being equal, the most promising object for further engagement is the one which exhibited the most recent progress. In our theoretical analysis, the size (as long as it is more than logarithmic in $n_x$) and the number of observed drops play only negligible roles compared to the recency of the last drop. This stands in contrast to empirical methods which often take the magnitude of the most recent progress as primary indicator (Baranes & Oudeyer, 2013; Kanitscheider et al., 2021; Gaven et al., 2025).

A caveat of the $P_x \leftrightarrow D_x$ correspondence is its use of the Busy Beaver function. Because BB grows faster than any computable function, the runtimes discussed here are mostly physically unrealizable. However, BB serves a vital theoretical role: it abstracts away machine dependence, ensuring these results reflect the *intrinsic* algorithmic nature of the objects rather than the specifics of a reference computer. This machine independence is important, since it is difficult to justify singling out any particular computer as the "most basic" or "most universal" (Müller, 2010). While the findings in this section are theoretical, they provide some conceptual grounding and suggest that "interestingness" can be a principled heuristic for navigating the space of computable objects. Section D in the Appendix tests the inductive property of interestingness experimentally in three different computationally universal paradigms and corroborates our results.

## 5. Alternative Views

We have already discussed the emergence of open-endedness from social interactions (Section 1), and the possibility of using LLMs to predict interestingness (Section 2). A valid critique of the *inductive property* of interesting-

ness is its apparent disregard for *content*. Perhaps a system should not predict future progress based solely on the shape of a complexity profile, but rather on the semantic nature of the object itself. From this perspective, an agent can recognize a task as interesting because it identifies familiar motifs—physics-like patterns, linguistic structures, or causal hierarchies—which tend to yield to further analysis. We argue, however, that accounting for the content is analogous to choosing what kind of prior to use. To an agent with no domain knowledge, the curve is the only universal signal available. But as a system matures, it develops meta models of compressibility—essentially domain-specific priors. These models allow the agent to classify strings into different "structural families." For instance, a string known to contain large amounts of random noise can prompt a prior algorithmically favoring objects that plateau early, while a string representing a mathematical proof might elicit a prior that favors deep objects and steady progress over many drops. Still, our analysis suggests that the complexity profile itself serves as a content-agnostic summary of an object's potential.

## 6. Conclusion & Future Work

We have argued that *Interestingness* is a necessary algorithmic heuristic for autonomous open-ended intelligence. By formalizing interestingness as a prospective assessment of future compression progress, we have grounded the concept in the framework of Algorithmic Information Theory. Our analysis of priors over computable objects—Length, Algorithmic, and Speed—reveals that the "inductive property of interestingness" is theoretically and empirically viable: past compression progress can indeed signal the potential for future discovery. With increasing stagnation length (gap since the last observed compression progress), the expected future compression progress diminishes exponentially. The exact inductivity properties, however, depend on the choice of prior.

This reliance on specific priors, in particular, warrants further investigation. An inversion of our inquiry could also be insightful: if we take the existence of inductive interestingness—the property that past compression progress reliably predicts future insight—as given, what must be true about the underlying distribution of objects? This is a fundamental question for recursive self-improvement: how must a "world" be structured to enable never-ending progress? Our previous analysis of universal priors suggests a somewhat limited memory; the probability of future progress is largely determined by the recency of the last drop. However, in structured domains, we intuitively expect that the *entire history* of progress matters. We do not expect long, steady sequences of discovery to terminate abruptly. This property of sustained, multi-level

learnability can be characterized as *scale-free* (Barabási & Albert, 1999) emergence. In algorithmic terms, drops in the $P_x$ profile represent transitions between different levels of description or emergence (Bédard & Bergeron, 2022). If an object exhibits scale-free emergence, its profile contains a steady continuation of such drops across many orders of magnitude of complexity. Natural phenomena—such as biological systems, or weather and climate—appear to possess this property: they provide an almost endless well of regularities where each discovery uncovers a new layer of puzzles. Also synthetic artifacts with similar characteristics exist: for example, certain fractals like the Mandelbrot set, or cellular automata like Conway's Game of Life. For such objects, we would expect that a long history of past compression progress is a more robust indicator of future learnability. While Jansma & Hoel (2025) have shown that such objects exhibiting causal contributions which are spread out across many levels of coarse-graining can be engineered, it remains a challenge to formally define the conditions required for a prior to prefer these scale-free structures.

Furthermore, we believe that to move beyond human-in-the-loop systems and towards truly self-improving open-ended intelligence, *introspective assessment* of our models is necessary. Current machine learning objectives focus almost exclusively on minimizing post-hoc loss. To achieve autonomy, we must develop introspective models capable of predicting their own learning progress. This requires architectures that do not just compress data, but explicitly model their own "compression frontier"—identifying where an increase in computational effort is likely to yield the highest gain. Our results suggest that one practical approach is to track how recently substantial progress was made on the data or tasks under consideration. An autonomous agent must have the ability to decide whether an object is boring (meaning additional runtime will not yield further compression) or interesting (meaning it holds the potential for a complexity drop given more reasoning tokens). Crucially, the exact nature of the computational effort in question requires a more rigorous taxonomy. Whether defined by longer recurrence, increased thinking steps, additional gradient descent iterations, or expanded model capacity, the underlying algorithmic relationship between these diverse resources remains a vital open question for future research. It remains to be seen to what extent complexity and runtime profiles could allow us to view these types of computational effort under the same lens, and taking recent progress in one domain also as an indicator for future potential in another.

By shifting our focus from pure learning to the principled selection of *what* to learn, we can begin to build systems that do not merely solve the tasks we give them, but autonomously seek to discover the richness of the universe they inhabit.

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

# A. Proposed Criteria for Interestingness

**Count-based, Space Coverage, and Maximum Entropy**    An object can be deemed interesting if it is *uncommon*—that is, if it has been encountered fewer times than other objects (Sutton, 1990; Bellemare et al., 2016; Tang et al., 2017). In high-dimensional spaces, this requires a density model to quantify novelty via smoothing. If an agent seeks to maximize the entropy of its state-visitation distribution, it will, in the limit, converge towards a policy that encounters all reachable objects with equal probability (Hazan et al., 2019; Mutti, 2023). In certain restricted settings, this is a reasonable measure: if it is possible for the cognitive system to encounter all reachable objects multiple times, uncommon objects yield the highest surprise in the Shannon sense. However, this does not naturally lead to open-ended learning; the system eventually converges towards a uniform occupancy and learning stagnates. Furthermore, if there are more possible objects than can ever be encountered, this measure of interestingness completely depends on the chosen form of coarse-graining or smoothing. In other words, it depends on a specific *model*.

**Prediction Error, Adversarial Criterion**    This notion moves beyond mere counting by introducing a predictive model. Here, interestingness is the inverse of the probability assigned by the model (the "adversarial" criterion). An object that is poorly predicted could be considered interesting (Schmidhuber, 1990; 1991a; Pathak et al., 2017). However, this fails to distinguish between epistemic uncertainty (reducible through learning) and aleatoric uncertainty (irreducible randomness). This is the "Noisy TV" problem (Schmidhuber, 2010): a subject using pure prediction error as a reward will get stuck observing a source of pure noise (like a static TV) because it is always unpredictable, yet provides zero learnable structure.

Common approaches to tackle this problem share the insight that an object loses its interestingness once nothing new can be learned about it.

**Information Gain, Bayesian Surprise**    If our model is set up in a probabilistic way, for example as a belief distribution over a set of hypotheses, then we can measure the information gain an object yields. Concretely, this is quantified as the Kullback-Leibler divergence between the posterior distribution, given the object, and the prior distribution. In broader terms, an object is interesting to the extent it leads to new insights and changes the model's "world view" (Storck et al., 1995; Itti & Baldi, 2005). Once nothing new can be learned from an object, the information gain subsides. Information gain can be measured not just over a set of fixed concrete hypotheses, but also over latent variables (Schmidhuber, 2003; Tishby & Zaslavsky, 2015; Sekar et al., 2020; Herrmann et al., 2025) or the predictions of existing data (Herrmann et al., 2026).

**Learning Progress, Competence Progress**    Closely related to information gain is the notion of learning progress (Schmidhuber, 1991b; Oudeyer et al., 2007; Stout & Barto, 2010). Instead of framing insight in terms of probabilistic inference and differences between posterior and prior, we can ask: Does the object lead to progress on some pre-defined objective function? An object is then interesting to the extent it improves the model's performance. Whether this is a reasonable measure of interestingness depends, of course, on the specific choice of objective function. Certainly not all tasks or objective functions are rich enough to allow open-ended learning. However, for many models—especially the ones we might want to use in the setups we mentioned, such as LLMs or world models (Schmidhuber, 2015; Ha & Schmidhuber, 2018; Bruce et al., 2024)—there is one dominant objective function: the negative log-likelihood (NLL). It can be argued that the success of the NLL as a training objective for models is due to the strong connection between learning and compression (Hutter, 2005; Delétang et al., 2024).

**Compression Progress**    This leads us to the concept of interestingness as compression progress (Schmidhuber, 2009; 2006). An object might be deemed interesting if it allows the model to better losslessly compress all available data. For a full compression of the data, we must account for the size of the model itself: what we measure is the number of bits required to describe the model in addition to the bits required to encode the data given the model. This two-part encoding of objects is the core idea behind the Minimum Description Length (MDL) principle (Wallace & Boulton, 1968; Rissanen, 1978). The compression progress associated with an object is thus the reduction of the total encoding size—the sum of the model complexity and the data residual—once the model has incorporated the object. A rich description of compute-bounded and observer-dependent MDL models has recently been presented by Finzi et al. (2026).

## B. How Profiles Relate to Current Machine Learning

Both types of profiles, $D_x$ and $P_x$, can be used to analyze important machine learning methods. The boundaries of the profiles correspond to optimal limits, which most real-world setups will not necessarily find. Still, we believe it is reasonable to assume that a well-calibrated system will stay close to the theoretical boundary, allowing the transfer of our theoretical analysis to machine learning practice.

**Neural Networks & Gradient Descent** Assume we train a neural network with Stochastic Gradient Descent (SGD) on some data $x$ using an NLL objective. We can plot the trajectory of training in the $(i, j)$ space of the $P_x$ profile, with the complexity of the weights on the x-axis and the log-size of the set of inputs assigned non-negligible probability on the y-axis. At random initialization, the network computes a near-uniform function over inputs—high log-set-size. The weights have initially very low complexity: All that is needed in order to describe the weights is the random seed and the random number generator. The *functional* complexity of this map is also low, since it can be described simply as "approximately uniform." As SGD proceeds, the function becomes more specific: log-set-size decreases as the model assigns higher probability to training data, and model complexity increases as the learned map encodes more structure, and the weights become more refined. If the network merely memorizes the data, this progression follows the trivial slope of $-1$: each additional bit of weight complexity eliminates roughly half the plausible candidates, with no net compression gain. A genuine structural discovery—for example the network finding a regularity that explains many examples at once—corresponds to a drop below this slope: description length falls faster than one bit per bit of added complexity. This is exactly a drop in the $P_x$ profile, and in MDL terms, it is the moment the two-part code (weight complexity + NLL) decreases. In practice, SGD instantiates a single trajectory through this space, not necessarily the optimal boundary of $P_x$, but the qualitative structure—periods of slope-$-1$ memorization punctuated by drops corresponding to genuine discoveries—should be preserved.

The same story can be told in the $D_x$ framing, where the connection to effort is more direct. Here the y-axis is total description length—weight complexity plus NLL—and the x-axis is runtime, meaning the number of gradient steps taken. Each drop in this profile represents a point where additional compute yielded a genuine compression gain rather than mere redistribution of bits between model and identifier.

**Chain-of-Thought and Recurrent Reasoning** Chain-of-Thought (CoT) reasoning maps naturally onto the $D_x$ profile. Each reasoning step extends the computation used to produce the final answer; thus, reasoning steps are counted on the x-axis. The y-axis represents the description length of the answer given the reasoning trace so far: how many bits are needed to specify the correct answer once the trace is fixed. The model complexity does not change during CoT reasoning; it simply provides a constant offset. A drop in this profile occurs where longer, more deliberate, more effortful reasoning leads to higher likelihood of the target. The same framing applies to depth-wise recurrent architectures, where each recurrence step plays the role of a reasoning token: runtime depth is the x-axis, and a drop corresponds to a recurrence step that increases the likelihood of the correct output.

**Model Scaling** Model size offers a third axis along which effort can be scaled. When models of increasing parameter count are each trained for a fixed number of SGD steps, they trace a trajectory in the $P_x$ profile: parameter count—after sufficient training—is a proxy for the complexity of the model (x-axis), and the NLL on training data is the log-size of the set of plausible inputs (y-axis). A drop in this profile occurs when a larger model finds a description of the data that is *shorter than the $-1$-slope extrapolation* from smaller models—that is, when increased capacity enables a genuine structural discovery rather than merely additional memorization.

Alternatively, the amount of training compute provides a more direct correspondence to the $D_x$ profile: Training FLOPs are counted on the x-axis and total description length—weight complexity plus NLL—on the y-axis. Here, a drop occurs when additional compute yields genuine compression of the full system, not merely redistribution between model size and loss. This total description length can be approximated empirically by the area under the training loss curve, via prequential coding (Blier & Ollivier, 2018): encoding each training example sequentially given the model at that step accumulates the per-step description lengths, approximating the MDL two-part cost.

## C. Mathematical Proofs

We start with some preliminaries. From Equations (2) and (3), we learn that the number of strings with a profile $O(C(P) + \log n_P)$-close to the given profile $P$ is

$$2^{k_P - m_P + O(\log n_P)}. \tag{7}$$

Here, we set $\epsilon$ to be a small constant value, which also justifies replacing $m_P(\epsilon)$, as defined by Vereshchagin & Shen (2016) for technicalities of their proof, with $m_P$.

A partial profile $\hat{P}$ characterizes sets containing a certain $x$, up to—but not including—complexity $t$. It is described by the values

$$\hat{n} := \min\{r \mid (0, r) \in \hat{P}\},$$
$$\hat{k} := \min\{r + s \mid (s, r) \in \hat{P}\}, \text{ and}$$
$$\hat{m} := \min\{r \mid (r, \hat{k} - r) \in \hat{P}\},$$

with $0 < \hat{m} < t \leq \hat{k} < \hat{n}$. We want to quantify how many strings there are for any possible continuation of the partial profile $\hat{P}$ beyond $t$ to a full profile $P \supset \hat{P}$ with $\{(i, j) \in P \,\forall i < t\} = \hat{P}$. For all these profiles $P$, $n_P = \hat{n}$, since the start of the profile is $\hat{P}$. For the continuations, we distinguish two different scenarios: Either there is no further drop after $\hat{P}$ (the blue trajectory in Figure 5), or there is at least one more drop (the green trajectories in the same figure). Let's start with the no-further-drop case, where progress *stops* before $t$. Here, the values $\hat{n}, \hat{k}$ and $\hat{m}$ characterize also the full profile. This means the number of strings that follow profile $\hat{P}$ with no further drops is, according to Equation 7,

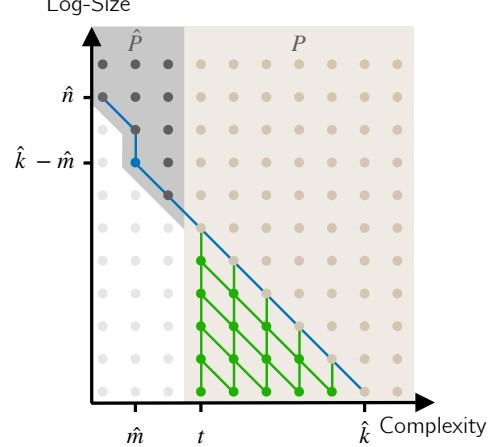

*Figure 5.* Continuation of profile $\hat{P}$ with no further drop (blue), and all possible continuations with $m \geq t$ (green).

$$N_{\text{stop}} := 2^{\hat{k} - \hat{m} + O(\log \hat{n})}.$$

Defining the constant $g_{\text{stop}} := 2^{O(\log \hat{n})}$, we write this as

$$N_{\text{stop}} = g_{\text{stop}} 2^{\hat{k} - \hat{m}}. \tag{8}$$

The constant $g_{\text{stop}}$ has a specific value, associated with the number of strings having a profile with the last drop set by $\hat{m}$ and $\hat{k}$.

If there is at least one more drop, the number of strings with a certain profile depends on the exact position of the last drop, characterized by $k$ and $m$. Since in this case, the complexity $m$ of the last drop is at least $t$, we describe further drop condition as "$m \geq t$". The total number of strings following profile $\hat{P}$ and then exhibiting a further drop is the sum of strings with a particular drop position, summed over all possible drop positions:

$$N_{\text{drop}} := \sum_{m=t}^{\hat{k}-1} \sum_{k=m}^{\hat{k}-1} 2^{k - m + O(\log \hat{n})}.$$

The outer sum enumerates the position of the drops along the complexity axis, the inner sum the size of the drop along the log-size axis. We again re-write this using a constant $g_{\text{drop}} := \sum_{m=t}^{\hat{k}-1} \sum_{k=m}^{\hat{k}-1} 2^{O(\log \hat{n})} = 2^{O(\log \hat{n})}$:

$$N_{\text{drop}} = \sum_{m=t}^{\hat{k}-1} \sum_{k=m}^{\hat{k}-1} 2^{O(\log \hat{n})} \sum_{m=t}^{\hat{k}-1} \sum_{k=m}^{\hat{k}-1} 2^{k-m}$$
$$= g_{\text{drop}} \sum_{m=t}^{\hat{k}-1} \sum_{k=m}^{\hat{k}-1} 2^{k-m}. \tag{9}$$

We have to consider a detail: Equation 7 does not quantify the number of strings with a profile of that matches exactly $P$, but the number of strings with a profile *close* to $P$. Hence, in the sums above, we may count the same string multiple times. However, the following Lemma shows that this double-counting does not contribute much.

**Lemma C.1.** *Consider the set of valid profiles with fixed $n$, and all allowed last drop positions parametrized by $(m, k)$-tuples. There are at most $O(k_x^2 \log^2 n)$ profiles from this set to which the profile $P_x$ of a given string $x$, with $n_x = n$, is $O(C(P) + O(\log n))$-close.*

*Proof.* Let $(m_x, k_x)$ be the parameters characterizing the last drop of $x$'s profile $P_x$. To be $O(C(P) + O(\log n))$-close to $P_x$, the last drop[2] of any profile $P$ has to occur within a square with a size of exactly that precision around the point $(m_x, k_x - m_x)$. There are only $O(C(P)^2 + O(\log n)^2)$ points in such a square. From the fact that $P$ can always be represented as a list of integers not bigger than $n$, it follows that $C(P) \leq \hat{k} \log n$. Therefore, the square of acceptable positions of the last drop contains $O(k_x^2 \log^2 n)$ points, each with their own valid profile. $\qquad\square$

The double-counting can thus be absorbed into the polynomial constant $g_{\text{drop}}$. Also this constant has a specific value, depending on the precise number of strings with each of the different last drops, and the mentioned double counting. We keep the explicit constants because they can be reduced: For example, $\frac{g_{\text{drop}}}{g_{\text{drop}}} = 1$, whereas in general $\frac{2^{O(\log \hat{n})}}{2^{O(\log \hat{n})}} = 2^{O(\log \hat{n}) - O(\log \hat{n})} = 2^{O(\log \hat{n})}$.

Each point in a profile corresponds to a two-part description of a string. The priors introduced in Section 4 give probabilistic weights to strings, depending on these descriptions.

## C.1. Length Prior

For the Length Prior $L$ (Equation 4), the probability of a string depends only on its length, i.e. on the value $\hat{n}$ of its profile $P$ (no matter if partial or non-partial, since the length is always determined from the start). Since all strings we are considering share the same initial partial profile $\hat{P}$, they have the same probability, and we can base our following calculations purely on the relative number of strings.

### C.1.1. PROOF OF PROPOSITION 4.1(I): PROBABILITY OF A FURTHER DROP

*Proof.* The probability that after $\hat{P}$ there will be a further drop is simply the fraction $\frac{N_{\text{drop}}}{N_{\text{drop}} + N_{\text{stop}}}$ of strings exhibiting a further drop. $N_{\text{drop}}$ (Equation 9) can be simplified:

$$
\begin{aligned}
N_{\text{drop}} &= g_{\text{drop}} \sum_{m=t}^{\hat{k}-1} \sum_{k=m}^{\hat{k}-1} 2^{k-m} \\
&= g_{\text{drop}} \left( 2^{\hat{k}-t+1} - \hat{k} + t - 2 \right) \\
&= 2^{O(\log \hat{n})} 2^{\hat{k}-t}
\end{aligned}
\tag{10}
$$

For $\hat{k} - t \gg 0$, we can approximate

$$
N_{\text{drop}} \approx g_{\text{drop}} 2^{\hat{k}-t+1}.
\tag{11}
$$

---

[2] The drop has to be of size $\Omega(C(P) + \log n)$. We are always assuming drop sizes that are more than logarithmic.

With that, we can calculate the probability of a further drop:

$$
\begin{aligned}
p_L(m \geq t \mid \hat{P}) &= \frac{N_{\text{drop}}}{N_{\text{drop}} + N_{\text{stop}}} \\
&= \frac{1}{1 + \frac{N_{\text{stop}}}{N_{\text{drop}}}} \\
&= \frac{1}{1 + \frac{2^{O(\log \hat{n})}}{2^{O(\log \hat{n})}} \frac{2^{\hat{k} - \hat{m}}}{2^{\hat{k} - t}}} \\
&= 2^{-(t - \hat{m}) + O(\log \hat{n})}
\end{aligned} \tag{12}
$$

It diminishes exponentially as the gap between the complexity of the last observed drop $\hat{m}$ and the observation cutoff $t$ grows. □

### C.1.2. PROBABILITY OF SPECIFIC DROP POSITION, GIVEN $m \geq t$

Given $\hat{P}$ and $m \geq t$, the number of strings with a profile where the last drop occurs at a specific complexity $m^*$ is

$$
\begin{aligned}
\sum_{m=t}^{\hat{k}-1} \sum_{k=m}^{\hat{k}-1} \mathbf{1}[m = m^*] 2^{k-m+O(\log \hat{n})} \\
= \sum_{k=m^*}^{\hat{k}-1} 2^{k-m^*+O(\log \hat{n})} \\
= 2^{\hat{k}-m^*+O(\log \hat{n})}.
\end{aligned} \tag{13}
$$

The probability of this drop position is hence

$$
\begin{aligned}
p_L(m = m^* \mid \hat{P}, m \geq t) &= \frac{2^{\hat{k}-m^*+O(\log \hat{n})}}{N_{\text{drop}}} \\
&= \frac{2^{\hat{k}-m^*+O(\log \hat{n})}}{2^{\hat{k}-t+O(\log \hat{n})}} \\
&= 2^{t-m^*+O(\log \hat{n})}.
\end{aligned} \tag{14}
$$

This shows that the probability of the last drop occurring at $m^*$ decays exponentially as $m^*$ moves away from the current threshold $t$. We now perform a parallel derivation for the final complexity $k^*$. The number of strings with a specific complexity $k^*$, given $\hat{P}$ and $m \geq t$, is

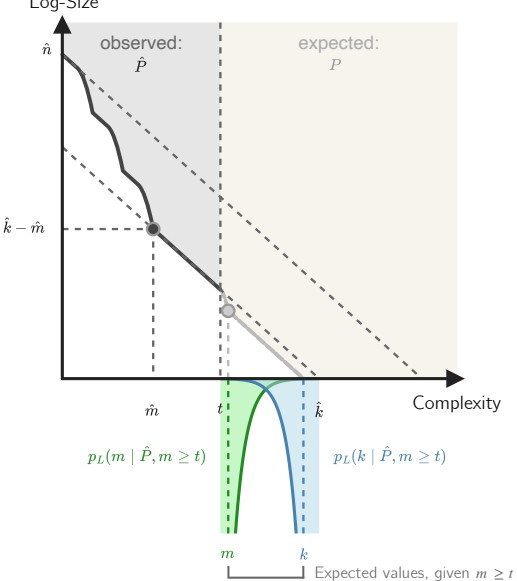

*Figure 6.* Expected continuation of profile $\hat{P}$ under the Length Prior, assuming the existence of a further drop.

$$
\begin{aligned}
\sum_{m=t}^{\hat{k}-1} \sum_{k=m}^{\hat{k}-1} \mathbf{1}[k = k^*] 2^{k-m+O(\log \hat{n})} &= \sum_{m=t}^{k^*} 2^{k^*-m+O(\log \hat{n})} \\
&= 2^{k^*-t+O(\log \hat{n})},
\end{aligned} \tag{15}
$$

and the probability of this complexity is

$$
\begin{aligned}
p_L(k = k^* \mid \hat{P}, m \geq t) &= \frac{2^{k^*-t+O(\log \hat{n})}}{N_{\text{drop}}} \\
&= \frac{2^{k^*-t+O(\log \hat{n})}}{2^{\hat{k}-t+O(\log \hat{n})}} \\
&= 2^{k^*-\hat{k}+O(\log \hat{n})}.
\end{aligned} \tag{16}
$$

These results are visualized in Figure 6.

### C.1.3. PROOF OF PROPOSITION 4.1(II): EXPECTED VALUES, GIVEN $m \geq t$

*Proof.* Now we calculate the expected values of $m$ and $k$ under the Length Prior, under the condition of a further drop $m \geq t$. For that, we first compute the sums of the number of strings with a certain profile, multiplied with the respective value of $m$ and $k$:

$$
\begin{aligned}
M_L &:= g_{\text{drop}} \sum_{m=t}^{\hat{k}-1} \sum_{k=m}^{\hat{k}-1} m 2^{k-m} \\
&= g_{\text{drop}} \left( 2^{\hat{k}-t+1} (t+1) - 2\hat{k} + \frac{1}{2} \left( -\hat{k}^2 - 3\hat{k} + t^2 - t - 4 \right) \right),
\end{aligned}
\tag{17}
$$

and

$$
\begin{aligned}
K_L &:= g_{\text{drop}} \sum_{m=t}^{\hat{k}-1} \sum_{k=m}^{\hat{k}-1} k 2^{k-m} \\
&= g_{\text{drop}} \left( 2^{\hat{k}-t+1} \left( \hat{k} - 2 \right) + \frac{1}{2} \left( -\hat{k}^2 + \hat{k} + t^2 - 5t + 8 \right) \right)
\end{aligned}
\tag{18}
$$

For $\hat{k} - t \gg 0$, these values simplify to

$$
M_L \approx g_{\text{drop}} 2^{\hat{k}-t+1} (t+1),
\tag{19}
$$

and

$$
K_L \approx g_{\text{drop}} 2^{\hat{k}-t+1} \left( \hat{k} - 2 \right).
\tag{20}
$$

The expected values are then, using Equation 11,

$$
\begin{aligned}
\mathbb{E}_{x \sim L}[m \mid \hat{P}, m \geq t] &= \frac{M_L}{N_{\text{drop}}} \\
&\approx \frac{g_{\text{drop}} 2^{\hat{k}-t+1} (t+1)}{g_{\text{drop}} 2^{\hat{k}-t+1}} \\
&= t + 1,
\end{aligned}
\tag{21}
$$

and

$$
\begin{aligned}
\mathbb{E}_{x \sim L}[k \mid \hat{P}, m \geq t] &= \frac{K_L}{N_{\text{drop}}} \\
&\approx \frac{g_{\text{drop}} 2^{\hat{k}-t+1} \left( \hat{k} - 2 \right)}{g_{\text{drop}} 2^{\hat{k}-t+1}} \\
&= \hat{k} - 2.
\end{aligned}
\tag{22}
$$

Under the Length Prior, further drops are most likely to occur close to the observation threshold $t$, and to be small, leading to a high singleton complexity. $\qquad \square$

C.1.4. PROOF OF PROPOSITION 4.1(III): EXPECTED PROGRESS, CONDITIONED ONLY ON $\hat{P}$

*Proof.* Observing the partial profile $\hat{P}$, the difference in complexity between the last observed drop at $\hat{m}$, and the expected last drop is

$$\mathbb{E}_{x \sim L}[m \mid \hat{P}] - \hat{m} = \hat{m}\left(1 - p_L(m \geq t \mid \hat{P})\right) + \mathbb{E}_{x \sim L}[m \mid \hat{P}, m \geq t]\, p_L(m \geq t \mid \hat{P}) - \hat{m}$$

$$= p_L(m \geq t \mid \hat{P})\left(\mathbb{E}_{x \sim L}[m \mid \hat{P}, m \geq t] - \hat{m}\right). \tag{23}$$

Since the second factor (the complexity difference) is at most linear in $\hat{k}$, it is dominated by the $2^{O(\log \hat{n})}$ of the probability, and thus, with Equations 12 and 21,

$$\mathbb{E}_{x \sim L}[m \mid \hat{P}] - \hat{m} = 2^{-(t - \hat{m}) + O(\log \hat{n})}. \tag{24}$$

The expected compression progress, relative to $\hat{k}$, is

$$\hat{k} - \mathbb{E}_{x \sim L}[k \mid \hat{P}] = \hat{k} - \hat{k}\left(1 - p_L(m \geq t \mid \hat{P})\right) - \mathbb{E}_{x \sim L}[k \mid \hat{P}, m \geq t]\, p_L(m \geq t \mid \hat{P})$$

$$= p_L(m \geq t \mid \hat{P})\left(\hat{k} - \mathbb{E}_{x \sim L}[k \mid \hat{P}, m \geq t]\right), \tag{25}$$

and again, with Equations 12 and 22,

$$\hat{k} - \mathbb{E}_{x \sim L}[k \mid \hat{P}] = 2^{-(t - \hat{m}) + O(\log \hat{n})}. \tag{26}$$

$\square$

C.1.5. PROBABILITY OF A DROP WITHIN A GIVEN WINDOW

We do not only care about the total future compression potential of a string, but perhaps even more about the near-term potential. To study this, we define the interval $d := \{t, \ldots, t + \Delta c - 1\}$, demarcating a specific window of interest (the green shaded area in Figure 7). We want to establish the probability of a drop happening within $d$, and the expected compression progress within the interval.

Since so far, we only cared about the indefinite future, we were able to only rely on the properties associated with the *last* drop. But now we care about *any* drop within $d$. This requires some additional care.

Apart from the case where the observed drop defined by $\hat{m}$ and $\hat{k}$ remains the last one, we distinguish three more scenarios:

- **Drop profiles (Green):** The last drop occurs within the window $d$.

- **Void profiles (Red):** The last drop occurs after $t + \Delta c$, but the drop is too shallow to allow any intermediate drops within $d$.

- **Perhaps profiles (Yellow):** The last drop occurs after the window, but the complexity gap is large enough that a preceding drop could exist within $d$.

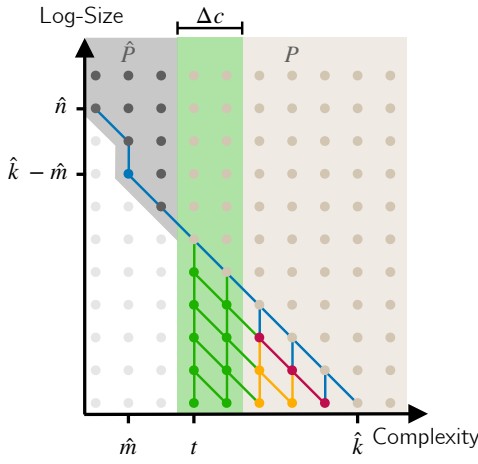

*Figure 7.* Continuation of profile $\hat{P}$. In blue, continuation with no further drop. In green, continuations with their last drop within $\Delta c$ after $t$. In red, continuations that have their last drop after the interval, and cannot have any drop within the interval. In yellow, continuations that have their last drop after the interval, but may have another drop within the interval.

The number of strings in the "drop" is, defining $g(d)_{\mathrm{drop}} := \sum_{m=t}^{t+\Delta c-1} \sum_{k=m}^{\hat{k}-1} 2^{O(\log \hat{n})} = 2^{O(\log \hat{n})}$

$$N(d)_{\mathrm{drop}} := g(d)_{\mathrm{drop}} \sum_{m=t}^{t+\Delta c-1} \sum_{k=m}^{\hat{k}-1} 2^{k-m}$$

$$= g(d)_{\mathrm{drop}} \left( 2^{\hat{k}-t+1} - 2^{\hat{k}-t-\Delta c+1} - \Delta c \right)$$

$$= 2^{\hat{k}-t+O(\log \hat{n})} - 2^{\hat{k}-t-\Delta c+O(\log \hat{n})}, \tag{27}$$

which for $\hat{k} - t \gg 0$ can be simplified to

$$N(d)_{\mathrm{drop}} \approx g(d)_{\mathrm{drop}} \left( 1 - 2^{-\Delta c} \right) 2^{\hat{k}-t}. \tag{28}$$

Defining $g(d)_{\mathrm{void}}$ and $g(d)_{\mathrm{perhaps}}$ equivalently, we get

$$N(d)_{\mathrm{void}} := g_{\mathrm{void}}(d) \sum_{m=t+\Delta c}^{\hat{k}-1} 2^{\hat{k}-1-m}$$

$$= g_{\mathrm{void}}(d) \left( 2^{\hat{k}-t-\Delta c} - 1 \right)$$

$$= 2^{\hat{k}-t-\Delta c+O(\log \hat{n})} \tag{29}$$

and

$$N(d)_{\mathrm{perhaps}} := g(d)_{\mathrm{perhaps}} \sum_{m=t+\Delta c}^{\hat{k}-2} \sum_{k=m}^{\hat{k}-2} 2^{k-m}$$

$$= g(d)_{\mathrm{perhaps}} \left( 2^{\hat{k}-t-\Delta c} + \Delta c - \hat{k} + t - 1 \right)$$

$$= 2^{\hat{k}-t-\Delta c+O(\log \hat{n})}. \tag{30}$$

Recall the value $N_{\mathrm{drop}}$ defined in Equation 9, that quantifies all strings with a drop after $\hat{P}$. The number of strings with a drop within $d$ is lower bounded by $N_{\mathrm{drop}} - (N(d)_{\mathrm{void}} + N(d)_{\mathrm{perhaps}})$ and upper bounded by $N_{\mathrm{drop}} - N(d)_{\mathrm{void}}$. This allows us to bound the probability:

$$\frac{N_{\mathrm{drop}} - (N(d)_{\mathrm{void}} + N(d)_{\mathrm{perhaps}})}{N_{\mathrm{drop}} + N_{\mathrm{stop}}} \le p_L(\text{drop within } d \mid \hat{P}) \le \frac{N_{\mathrm{drop}} - N(d)_{\mathrm{void}}}{N_{\mathrm{drop}} + N_{\mathrm{stop}}} \tag{31}$$

Notice that, due to the $O(\log \hat{n})$-term in the exponent,

$$N(d)_{\mathrm{void}} + N(d)_{\mathrm{perhaps}} = 2^{\hat{k}-t-\Delta c+O(\log \hat{n})} = N(d)_{\mathrm{void}}. \tag{32}$$

Hence,

$$p_L(\text{drop within } d \mid \hat{P}) = \frac{N_{\mathrm{drop}} - N(d)_{\mathrm{void}}}{N_{\mathrm{drop}} + N_{\mathrm{stop}}}$$

$$= \frac{2^{O(\log \hat{n})} 2^{\hat{k}-t} - 2^{O(\log \hat{n})} 2^{\hat{k}-t-\Delta c}}{2^{O(\log \hat{n})} 2^{\hat{k}-t} + 2^{O(\log \hat{n})} 2^{\hat{k}-\hat{m}}}$$

$$= 2^{\hat{m}-t+O(\log \hat{n})} \left( 1 - 2^{-\Delta c} \right). \tag{33}$$

We can see that the probability of a further drop is highly concentrated close to $t$, and intervals of larger size $\Delta c$ increase the probability by an exponentially vanishing amount. Furthermore, since the probability of the last drop occurring within $d$ is $p_L(\text{last drop within } d \mid \hat{P}) = \frac{N_{\mathrm{drop}} - (N(d)_{\mathrm{void}} + N(d)_{\mathrm{perhaps}})}{N_{\mathrm{drop}} + N_{\mathrm{stop}}}$, it follows that, up to the usual logarithmic precision in the exponents,

$$p_L(\text{last drop within } d \mid \hat{P}) = p_L(\text{drop within } d \mid \hat{P}). \tag{34}$$

This means that the profiles with the last drop within $d$ dominate the probability of having *any* drop within $d$.

### C.1.6. EXPECTED PROGRESS WITHIN A GIVEN WINDOW

Similarly to Sections C.1.3 and C.1.4, we can calculate the expected compression progress within $d$. Again, we start with the expected value for $k$, given that the last drop is within the interval $d$. The weighted number of corresponding strings is

$$
K_L(d) := g(d)_{\text{drop}} \sum_{m=t}^{t+\Delta c-1} \sum_{k=m}^{\hat{k}-1} k 2^{k-m}
$$

$$
= g(d)_{\text{drop}}(\hat{k}-2)(2^{\hat{k}-t+1} - 2^{\hat{k}-t-\Delta c+1}) + g(d)_{\text{drop}} \frac{\Delta c}{2}(5 - 2t - \Delta c), \tag{35}
$$

$$
K_L(d) \approx g(d)_{\text{drop}}(\hat{k}-2)\left(1 - 2^{-\Delta c}\right) 2^{\hat{k}-t}. \tag{36}
$$

With that, the expected value for $k$, assuming that the last drop happens within $d$ and using Equation 28, is

$$
\mathbb{E}_{x \sim L}[k \mid \hat{P}, \text{ last drop within } d] = \frac{K_L(d)}{N(d)_{\text{drop}}} \approx \hat{k} - 2. \tag{37}
$$

From Equation 34 it follows that also

$$
\mathbb{E}_{x \sim L}[k \mid \hat{P}, \text{ drop within } d] \approx \hat{k} - 2.
$$

Now we can compute the expected compression progress within $d$. Let $k_{<t+\Delta c}$ be the complexity of the last drop before $t + \Delta c$ (but not necessarily the last drop of the whole profile). That means, $k_{<t+\Delta c}$ represents the "achieved complexity" within the window, regardless of whether the string eventually becomes even simpler later in the profile. Then,

$$
\hat{k} - \mathbb{E}_{x \sim L}[k_{<t+\Delta c} \mid \hat{P}] = p_L(\text{drop within } d \mid \hat{P})\left(\hat{k} - \mathbb{E}_{x \sim L}[k \mid \hat{P}, \text{ drop within } d]\right)
$$

$$
\approx 2^{\hat{m}-t+O(\log \hat{n})}\left(1 - 2^{-\Delta c}\right). \tag{38}
$$

### C.2. Algorithmic Prior

The Length Prior from the previous section has the property that all strings of length $\hat{n}$ have the same probability. This is not the case for the Algorithmic Prior $M$ (Equation 5): Here, the probability of a string is determined by its complexity, i.e. the value $k$ of its profile $P$. Nevertheless, we can proceed in a manner similar to before.

### C.2.1. PROOF OF PROPOSITION 4.2(I): PROBABILITY OF FURTHER DROP

*Proof.* Instead of simply counting strings, we weigh each string with its prior probability. The unnormalized statistical weight of profiles with no further drop under the Algorithmic Prior $M$ is

$$
Z_{\text{stop}}^M := 2^{\hat{k}-\hat{m}+O(\log \hat{n})} 2^{-\hat{k}}
$$

$$
= 2^{O(\log \hat{n})} 2^{-\hat{m}}
$$

$$
= g_{\text{stop}} 2^{-\hat{m}}. \tag{39}
$$

*With* at least one more drop, the weighted number of strings is

$$
Z_{\text{drop}}^M := \sum_{m=t}^{\hat{k}-1} \sum_{k=m}^{\hat{k}-1} 2^{k-m+O(\log \hat{n})} 2^{-k}
$$

$$
= g_{\text{drop}} \sum_{m=t}^{\hat{k}-1} \sum_{k=m}^{\hat{k}-1} 2^{-m}
$$

$$
= g_{\text{drop}}\left(\left(\hat{k} - t - 1\right) 2^{1-t} + 2^{1-\hat{k}}\right)
$$

$$
= 2^{-t+O(\log \hat{n})} + 2^{-\hat{k}+O(\log \hat{n})} \tag{40}
$$

For $\hat{k} \gg t$, this is

$$Z_{\text{drop}}^M \approx g_{\text{drop}}\left(\hat{k} - t - 1\right) 2^{1-t}. \tag{41}$$

This yields the probability of a further drop

$$
\begin{aligned}
p_M(m \geq t \mid \hat{P}) &= \frac{1}{1 + \frac{Z_{\text{stop}}^M}{Z_{\text{drop}}^M}} \\
&= \frac{1}{1 + \frac{2^{O(\log \hat{n})}}{2^{O(\log \hat{n})}} \frac{2^{-\hat{m}}}{\left(\left(\hat{k}-t-1\right)2^{1-t}+2^{1-\hat{k}}\right)}} \\
&= \frac{1}{1 + \frac{2^{-\hat{m}+O(\log \hat{n})}}{2^{-t+O(\log \hat{n})}+2^{-\hat{k}+O(\log \hat{n})}}}.
\end{aligned} \tag{42}
$$

Here, we have a dependence on $\hat{k}$ in the denominator, which does not exist for the Length Prior (compare Equation 12). However, this dependence vanishes for $\hat{k} - t \gg 0$:

$$
\begin{aligned}
p_M(m \geq t \mid \hat{P}) &= \frac{1}{1 + \frac{Z_{\text{stop}}^M}{Z_{\text{drop}}^M}} \\
&\approx \frac{1}{1 + \frac{2^{-\hat{m}+O(\log \hat{n})}}{2^{-t+O(\log \hat{n})}}} \\
&= 2^{-(t-\hat{m})+O(\log \hat{n})}
\end{aligned} \tag{43}
$$

$\square$

### C.2.2. PROBABILITY OF SPECIFIC DROP POSITION, GIVEN $m \geq t$

Given $\hat{P}$ and $m \geq t$, the number of strings with a profile where the last drop occurs at a specific complexity $m^*$, weighted by the Algorithmic Prior, is:

$$
\begin{aligned}
\sum_{m=t}^{\hat{k}-1} \sum_{k=m}^{\hat{k}-1} \mathbf{1}[m = m^*] 2^{k-m+O(\log \hat{n})} 2^{-k} \\
= \sum_{k=m^*}^{\hat{k}-1} 2^{-m^*+O(\log \hat{n})} \\
= 2^{-m^*+O(\log \hat{n})}.
\end{aligned} \tag{44}
$$

The probability of this drop position is hence:

$$
\begin{aligned}
p_M(m = m^* \mid \hat{P}, m \geq t) &= \frac{2^{-m^*+O(\log \hat{n})}}{Z_{\text{drop}}^M} \\
&= \frac{2^{-m^*+O(\log \hat{n})}}{2^{-t+O(\log \hat{n})} + 2^{-\hat{k}+O(\log \hat{n})}}.
\end{aligned} \tag{45}
$$

For $\hat{k} - t \gg 0$, this simplifies to

$$p_M(m = m^* \mid \hat{P}, m \geq t) \approx 2^{t-m^*+O(\log \hat{n})}, \tag{46}$$

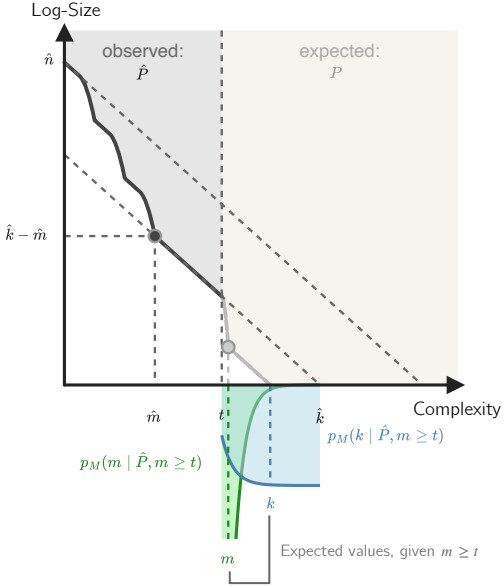

*Figure 8.* Expected continuation of profile $\hat{P}$ under the Algorithmic Prior, assuming the existence of a further drop.

which matches the Length Prior result in Equation 14.

In contrast, the weighted number of strings with a specific complexity $k^*$, given $\hat{P}$ and $m \geq t$, is:

$$2^{O(\log \hat{n})} \sum_{m=t}^{\hat{k}-1} \sum_{k=m}^{\hat{k}-1} \mathbf{1}[k = k^*] 2^{k-m} 2^{-k}$$
$$= 2^{O(\log \hat{n})} \sum_{m=t}^{k^*} 2^{-m}$$
$$= 2^{-t+O(\log \hat{n})} - 2^{-k^*+O(\log \hat{n})}. \tag{47}$$

The probability of this complexity is:

$$p_M(k = k^* \mid \hat{P}, m \geq t) = \frac{2^{-t+O(\log \hat{n})} - 2^{-k^*+O(\log \hat{n})}}{2^{-t+O(\log \hat{n})} + 2^{-\hat{k}+O(\log \hat{n})}}. \tag{48}$$

For $\hat{k} - t \gg 0$, this yields:

$$p_M(k = k^* \mid \hat{P}, m \geq t) \approx 1 - 2^{t-k^*+O(\log \hat{n})}. \tag{49}$$

Unlike the Length Prior (Equation 16, Figure 6), where the distribution of $k$ vanishes exponentially as it moves away from $\hat{k}$, the Algorithmic Prior results in a distribution that is nearly uniform for $k^* \gg t$ (see Figure 8).

C.2.3. PROOF OF PROPOSITION 4.2(II): EXPECTED VALUES, GIVEN $m \geq t$

Probabilistically weighted sums of the number of strings with a certain profile, multiplied with the respective value of $m$ and $k$:

$$M_M := g_{\text{drop}} \sum_{m=t}^{\hat{k}-1} \sum_{k=m}^{\hat{k}-1} m 2^{-m}$$
$$= g_{\text{drop}} \left( \left( (\hat{k} - t - 1)(2t + 2) - 4 \right) 2^{-t} + (2\hat{k} + 6) 2^{-\hat{k}} \right) \tag{50}$$
$$K_M := g_{\text{drop}} \sum_{m=t}^{\hat{k}-1} \sum_{k=m}^{\hat{k}-1} k 2^{-m}$$
$$= g_{\text{drop}} \left( \left( (\hat{k} - t - 1)(\hat{k} + t) - 2 \right) 2^{-t} + (2\hat{k} + 2) 2^{-\hat{k}} \right) \tag{51}$$

For $\hat{k} - t \gg 0$, these values simplify to

$$M_M \approx g_{\text{drop}}(\hat{k} - t - 1)(2t + 2) 2^{-t}, \tag{52}$$
$$K_M \approx g_{\text{drop}}(\hat{k} - t - 1)(\hat{k} + t) 2^{-t}. \tag{53}$$

The expected values are then, using Equation 41

$$\mathbb{E}_{x \sim M}[m \mid \hat{P}, m \geq t] = \frac{M_M}{Z_{\text{drop}}^M}$$

$$\approx \frac{g_{\text{drop}}(\hat{k} - t - 1)(2t + 2)2^{-t}}{g_{\text{drop}}\left(\hat{k} - t - 1\right)2^{1-t}}$$

$$= t + 1, \tag{54}$$

$$\mathbb{E}_{x \sim M}[k \mid \hat{P}, m \geq t] = \frac{K_M}{Z_{\text{drop}}^M}$$

$$\approx \frac{g_{\text{drop}}(\hat{k} - t - 1)(\hat{k} + t)2^{-t}}{g_{\text{drop}}\left(\hat{k} - t - 1\right)2^{1-t}}$$

$$= \frac{\hat{k} + t}{2}. \tag{55}$$

Here, the approximate uniformity from Equation 49 comes to bear: The expectation of $k$ is significantly lower compared to the Length Prior (compare Equation 22).

### C.2.4. PROOF OF PROPOSITION 4.2 (III): EXPECTED PROGRESS, CONDITIONED ONLY ON $\hat{P}$

*Proof.* Nevertheless, just as with the Length Prior, the expected progress, conditioned only on $\hat{P}$ is dominated by the drop probability:

$$\mathbb{E}_{x \sim M}[m \mid \hat{P}] - \hat{m} \approx 2^{-(t-\hat{m})+O(\log \hat{n})}, \tag{56}$$

and

$$\hat{k} - \mathbb{E}_{x \sim M}[k \mid \hat{P}] \approx 2^{-(t-\hat{m})+O(\log \hat{n})}. \tag{57}$$

$\square$

### C.2.5. PROBABILITY OF A DROP WITHIN A GIVEN WINDOW

Analogously to the Length Prior (Section C.1.5), we calculate the weight of the three cases:

$$Z(d)_{\text{drop}}^M := g(d)_{\text{drop}} \sum_{m=t}^{t+\Delta c-1} \sum_{k=m}^{\hat{k}-1} 2^{-m}$$

$$= g(d)_{\text{drop}} 2^{-t+1} \left(\left(1 - 2^{-\Delta c}\right)(\hat{k} - t - 1) + \Delta c 2^{-\Delta c}\right)$$

$$= 2^{-t+O(\log \hat{n})} - 2^{-t-\Delta c+O(\log \hat{n})} \tag{58}$$

$$Z(d)_{\text{drop}}^M \approx g(d)_{\text{drop}} 2^{-t+1}\left(1 - 2^{-\Delta c}\right)(\hat{k} - t) \tag{59}$$

$$Z(d)_{\text{void}}^M := g_{\text{void}}(d) \sum_{m=t+\Delta c}^{\hat{k}-1} 2^{-m-1}$$

$$= g_{\text{void}}(d)\left(2^{-t-\Delta c} - 2^{-\hat{k}}\right)$$

$$= 2^{-t-\Delta c+O(\log \hat{n})} - 2^{-\hat{k}+O(\log \hat{n})} \tag{60}$$

$$Z(d)_{\text{perhaps}}^M := g(d)_{\text{perhaps}} \sum_{m=t+\Delta c}^{\hat{k}-2} \sum_{k=m}^{\hat{k}-2} 2^{-m}$$

$$= g(d)_{\text{perhaps}} \left( (\hat{k} - t - \Delta c - 2)2^{-t-\Delta c+1} + 2^{-\hat{k}+2} \right)$$

$$= 2^{-t-\Delta c+O(\log \hat{n})} + 2^{-\hat{k}+O(\log \hat{n})} \tag{61}$$

Using Equation 41, we can bound the probability:

$$\frac{Z_{\text{drop}}^M - \left( Z(d)_{\text{void}}^M + Z(d)_{\text{perhaps}}^M \right)}{Z_{\text{drop}}^M + Z_{\text{stop}}^M} \le p_M(\text{drop within } d \mid \hat{P}) \le \frac{Z_{\text{drop}}^M - Z(d)_{\text{void}}^M}{Z_{\text{drop}}^M + Z_{\text{stop}}^M} \tag{62}$$

And again, due to the $O(\log \hat{n})$-term in the exponent,

$$Z(d)_{\text{void}}^M + Z(d)_{\text{perhaps}}^M = 2^{-t-\Delta c+O(\log \hat{n})} - 2^{-\hat{k}+O(\log \hat{n})} = Z(d)_{\text{void}}^M. \tag{63}$$

Hence,

$$p_M(\text{drop within } d \mid \hat{P}) = \frac{Z_{\text{drop}}^M - Z(d)_{\text{void}}^M}{Z_{\text{drop}}^M + Z_{\text{stop}}^M}$$

$$= 2^{O(\log \hat{n})} \frac{2^{-t} + 2^{-\hat{k}} - 2^{-t-\Delta c} + 2^{-\hat{k}}}{2^{-t} + 2^{-\hat{k}} + 2^{-\hat{m}}}$$

$$= \frac{2^{-t+O(\log \hat{n})} \left(1 - 2^{-\Delta c}\right) + 2^{-\hat{k}+O(\log \hat{n})}}{2^{O(\log \hat{n})} \left(2^{-t} + 2^{-\hat{k}} + 2^{-\hat{m}}\right)}, \tag{64}$$

and for $\hat{k} - t \gg 0$,

$$p_M(\text{drop within } d \mid \hat{P}) \approx \frac{2^{-t+O(\log \hat{n})} \left(1 - 2^{-\Delta c}\right)}{2^{-t+O(\log \hat{n})} + 2^{-\hat{m}+O(\log \hat{n})}}$$

$$= 2^{-(t-\hat{m})+O(\log \hat{n})} \left(1 - 2^{-\Delta c}\right). \tag{65}$$

Also here, larger values of $\Delta c$ contribute only by an exponentially vanishing amount. It again follows that

$$p_M(\text{last drop within } d \mid \hat{P}) = p_M(\text{drop within } d \mid \hat{P}). \tag{66}$$

### C.2.6. EXPECTED PROGRESS WITHIN A GIVEN WINDOW

Same procedure as Section C.1.6: The weighted number of corresponding strings is

$$K_M(d) := g(d)_{\text{drop}} \sum_{m=t}^{t+\Delta c-1} \sum_{k=m}^{\hat{k}-1} k 2^{-m}$$

$$= g(d)_{\text{drop}} 2^{-t} \left( \left(1 - 2^{-\Delta c}\right) (\hat{k} + t)(\hat{k} - t - 1) + 2^{-\Delta c}(\Delta c + 2)(\Delta c + 1) - 2 \right),$$

$$K_M(d) \approx g(d)_{\text{drop}} 2^{-t} \left(1 - 2^{-\Delta c}\right) (\hat{k} + t)(\hat{k} - t). \tag{67}$$

The expected value for $k$, assuming that the last drop happens within $d$ and using Equation 59, is

$$\mathbb{E}_{x \sim M}[k \mid \hat{P}, \text{ last drop within } d] = \frac{K_M(d)}{Z(d)_{\text{drop}}^M} \approx \frac{\hat{k} + t}{2}. \tag{68}$$

From Equation 66 it follows that

$$\mathbb{E}_{x \sim M}[k \mid \hat{P}, \text{ drop within } d] \approx \frac{\hat{k} + t}{2},$$ (69)

and

$$\hat{k} - \mathbb{E}_{x \sim M}[k_{<t+\Delta c} \mid \hat{P}] = p_M(\text{drop within } d \mid \hat{P}) \left( \hat{k} - \mathbb{E}_{x \sim M}[k \mid \hat{P}, \text{ drop within } d] \right)$$
$$\approx 2^{-(t-\hat{m})+O(\log \hat{n})} \left( 1 - 2^{-\Delta c} \right).$$ (70)

## C.3. Comparison Between Length and Algorithmic Prior

The analytic forms of the further drop probability (Equations 12 and 43), as well as the expected progress (Equations 26 and 57) are identical for Length and Algorithmic Prior. However, we can show that there is in fact a difference between the two priors.

### C.3.1. PROOF OF PROPOSITION 4.3(I): RATIO OF DROP PROBABILITIES

*Proof.* Using Equations 8, 11, 39, and 41, we can calculate the ratio of the drop probabilities:

$$\frac{p_M(m \geq t \mid \hat{P})}{p_L(m \geq t \mid \hat{P})} = \frac{Z_{\text{drop}}^M (N_{\text{drop}} + N_{\text{stop}})}{N_{\text{drop}} \left( Z_{\text{drop}}^M + Z_{\text{stop}}^M \right)}$$

$$= \frac{g_{\text{drop}} \left( \hat{k} - t - 1 \right) 2^{1-t} \left( g_{\text{drop}} 2^{\hat{k}-t+1} + g_{\text{stop}} 2^{\hat{k}-\hat{m}} \right)}{g_{\text{drop}} 2^{\hat{k}-t+1} \left( g_{\text{drop}} \left( \hat{k} - t - 1 \right) 2^{1-t} + g_{\text{stop}} 2^{-\hat{m}} \right)}$$

$$= 1 + \frac{\left( \hat{k} - t - 2 \right) g_{\text{stop}} 2^{\hat{k}-t-\hat{m}+1}}{g_{\text{drop}} \left( \hat{k} - t - 1 \right) 2^{\hat{k}-2t+2} + g_{\text{stop}} 2^{\hat{k}-t-\hat{m}+1}}$$

$$= 1 + \frac{\hat{k} - t - 2}{1 + \frac{g_{\text{drop}}}{g_{\text{stop}}} \left( \hat{k} - t - 1 \right) 2^{-(t-\hat{m}-1)}}$$

$$= 1 + \frac{\hat{k} - t - 2}{1 + 2^{-(t-\hat{m})+O(\log \hat{n})}}$$ (71)

For $t - \hat{m} \gg \log \hat{n}$, this simplifies to

$$\frac{p_M(m \geq t \mid \hat{P})}{p_L(m \geq t \mid \hat{P})} \approx \hat{k} - t - 1.$$ (72)

This result implies that the future drop probability under the Algorithmic Prior is higher than under the Length Prior by a factor approximately linear in the complexity gap $\hat{k} - t$. □

### C.3.2. PROOF OF PROPOSITION 4.3(II): RATIO OF COMPRESSION PROGRESS

*Proof.* The ratio of expected compression progress is derived by combining the ratio of discovery probabilities with the ratio of conditional gains:

$$\frac{\hat{k} - \mathbb{E}_{x \sim M}[k \mid \hat{P}]}{\hat{k} - \mathbb{E}_{x \sim L}[k \mid \hat{P}]} = \frac{p_M(m \geq t \mid \hat{P}) \left( \hat{k} - \mathbb{E}_{x \sim M}[k \mid \hat{P}, m \geq t] \right)}{p_L(m \geq t \mid \hat{P}) \left( \hat{k} - \mathbb{E}_{x \sim L}[k \mid \hat{P}, m \geq t] \right)}.$$ (73)

Assuming $\hat{k} - t \gg 0$ and $t - \hat{m} \gg \log \hat{n}$, we substitute the results from Equations 22, 55, and 72:

$$\frac{\hat{k} - \mathbb{E}_{x \sim M}[k \mid \hat{P}]}{\hat{k} - \mathbb{E}_{x \sim L}[k \mid \hat{P}]} \approx \left(\hat{k} - t - 1\right) \frac{\hat{k} - \frac{1}{2}\left(\hat{k} + t\right)}{\hat{k} - \left(\hat{k} - 2\right)}$$

$$= \frac{1}{4}\left(\hat{k} - t - 1\right)\left(\hat{k} - t\right). \tag{74}$$

Thus, the expected compression progress under the Algorithmic Prior is larger than under the Length Prior by a factor that is approximately quadratic in the remaining complexity gap $\hat{k} - t$. $\qquad\square$

### C.4. Speed Prior

The Speed Prior (Equation 6) assesses the probability of a string based on both the length and the log runtime of programs computing that string. Using the $P_x \leftrightarrow D_x$ correspondence (Equation 1), we can convert the value $m$ into a (extremely fast-growing) runtime, which is punished by the Speed Prior. For any sufficiently large value $i$, $\log \mathrm{BB}(i)/\log \mathrm{BB}(i+1) \approx 0$.

#### C.4.1. PROOF OF PROPOSITION 4.4(I): PROBABILITY OF FURTHER DROP

*Proof.* Unnormalized statistical weight of profiles with no further drop under the Speed Prior $S$:

$$Z_{\mathrm{stop}}^{S} := 2^{\hat{k} - \hat{m} + O(\log \hat{n})} 2^{-\hat{k} - \log \mathrm{BB}(\hat{m})}$$

$$= g_{\mathrm{stop}} 2^{-\hat{m} - \log \mathrm{BB}(\hat{m})} \tag{75}$$

For continuations with at least one more drop, the weight is:

$$Z_{\mathrm{drop}}^{S} := \sum_{m=t}^{\hat{k}-1} \sum_{k=m}^{\hat{k}-1} 2^{k - m + O(\log \hat{n})} 2^{-k - \log \mathrm{BB}(m)}$$

$$\approx g_{\mathrm{drop}} \sum_{k=t}^{\hat{k}-1} 2^{-t - \log \mathrm{BB}(t)}$$

$$= g_{\mathrm{drop}} (\hat{k} - t) 2^{-t - \log \mathrm{BB}(t)}, \tag{76}$$

where the approximation follows from the fact that the $m = t$ term dominates due to the extreme growth of BB. This yields the probability of a further drop

$$p_S(m \geq t \mid \hat{P}) = \frac{Z_{\mathrm{drop}}^{S}}{Z_{\mathrm{drop}}^{S} + Z_{\mathrm{stop}}^{S}}$$

$$\approx \frac{g_{\mathrm{drop}}(\hat{k} - t) 2^{-t - \log \mathrm{BB}(t)}}{g_{\mathrm{drop}}(\hat{k} - t) 2^{-t - \log \mathrm{BB}(t)} + g_{\mathrm{stop}} 2^{-\hat{m} - \log \mathrm{BB}(\hat{m})}}$$

$$\approx 0, \tag{77}$$

since $t > \hat{m}$. $\qquad\square$

#### C.4.2. PROBABILITIES OF SPECIFIC DROP POSITIONS GIVEN A FURTHER DROP

The probability of this drop position, using Equation 76, is hence

$$p_S(m = m^* \mid \hat{P}, m \geq t) = \frac{2^{-m^* - \log \mathrm{BB}(m^*) + O(\log \hat{n})}}{Z_{\mathrm{drop}}^{S}}$$

$$\approx \begin{cases} 1 & \text{if } m^* = t \\ 0 & \text{otherwise.} \end{cases} \tag{78}$$

Given $\hat{P}$ and $m \geq t$, we evaluate the weighted number of strings where the last drop occurs at complexity $m^*$:

$$\sum_{m=t}^{\hat{k}-1} \sum_{k=m}^{\hat{k}-1} \mathbf{1}[m = m^*] 2^{k-m+O(\log \hat{n})} 2^{-k-\log \mathrm{BB}(m)}$$

$$= \sum_{k=m^*}^{\hat{k}-1} 2^{-m^*-\log \mathrm{BB}(m^*)+O(\log \hat{n})}$$

$$= 2^{-m^*-\log \mathrm{BB}(m^*)+O(\log \hat{n})}. \tag{79}$$

Similarly, the number of strings with a specific complexity $k^*$, given $\hat{P}$ and $m \geq t$, is

$$\sum_{m=t}^{\hat{k}-1} \sum_{k=m}^{\hat{k}-1} \mathbf{1}[k = k^*] 2^{k-m+O(\log \hat{n})} 2^{-k-\log \mathrm{BB}(m)}$$

$$= \sum_{m=t}^{k^*} 2^{-m-\log \mathrm{BB}(m)+O(\log \hat{n})}$$

$$= 2^{-t-\log \mathrm{BB}(t)+O(\log \hat{n})}. \tag{80}$$

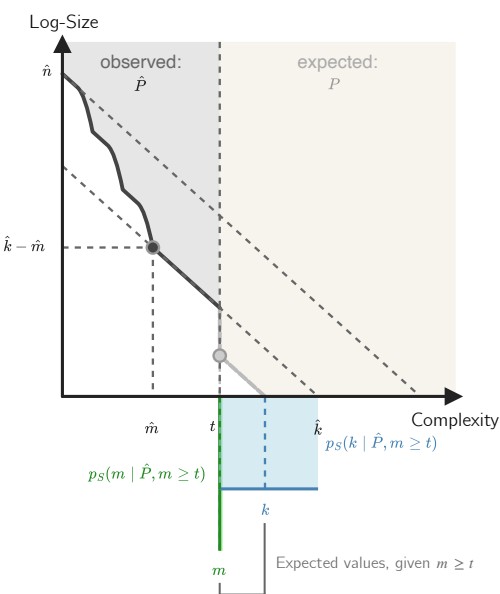

*Figure 9.* Expected continuation of profile $\hat{P}$ under the Speed Prior, assuming the existence of a further drop.

The resulting probability distribution for $k^*$ is approximately constant across the available range (see Figure 9):

$$p_S(k = k^* \mid \hat{P}, m \geq t) = \frac{2^{-t-\log \mathrm{BB}(t)+O(\log \hat{n})}}{Z_{\mathrm{drop}}^S} = 2^{O(\log \hat{n})}. \tag{81}$$

### C.4.3. PROOF OF PROPOSITION 4.4(II) & (III): EXPECTED VALUES

*Proof.* Probabilistically weighted sums of the number of strings with a certain profile, multiplied with the respective value of $m$ and $k$:

$$M_S := g_{\mathrm{drop}} \sum_{m=t}^{\hat{k}-1} \sum_{k=m}^{\hat{k}-1} m 2^{-m-\log \mathrm{BB}(m)}$$

$$\approx g_{\mathrm{drop}} \sum_{k=t}^{\hat{k}-1} t 2^{-t-\log \mathrm{BB}(t)}$$

$$= g_{\mathrm{drop}} \, t(\hat{k} - t) 2^{-t-\log \mathrm{BB}(t)} \tag{82}$$

$$K_S := g_{\mathrm{drop}} \sum_{m=t}^{\hat{k}-1} \sum_{k=m}^{\hat{k}-1} k 2^{-m-\log \mathrm{BB}(m)}$$

$$\approx g_{\mathrm{drop}} \sum_{k=t}^{\hat{k}-1} k 2^{-t-\log \mathrm{BB}(t)}$$

$$= g_{\mathrm{drop}} (\hat{k} - t)(\hat{k} + t - 1) 2^{-t-1-\log \mathrm{BB}(t)} \tag{83}$$

The expected values are then

$$\mathbb{E}[m \mid \hat{P}, m \geq t] \approx \frac{M_S}{Z_{\mathrm{drop}}^S} = t, \text{ and} \tag{84}$$

$$\mathbb{E}[k \mid \hat{P}, m \geq t] \approx \frac{K_S}{Z_{\mathrm{drop}}^S} = \frac{\hat{k} + t - 1}{2}. \tag{85}$$

Since the probability of a further drop $p_S(m \geq t \mid \hat{P}) \approx 0$, the unconditioned expectations remain at their observed values:

$$\mathbb{E}[m \mid \hat{P}] \approx \hat{m}, \text{ and } \mathbb{E}[k \mid \hat{P}] \approx \hat{k}. \tag{86}$$

Consequently, the expected compression progress is $0$. $\qquad \square$

Following this same logic for a finite window $d$, both the drop probability and the expected progress vanish:

$$p_S(\text{drop within } d \mid \hat{P}) \approx 0, \text{ and } \mathbb{E}_{x \sim S}[k_{<t+\Delta c} \mid \hat{P}] \approx \hat{k}. \tag{87}$$

# D. Empirical Results

The theoretical results from the previous section hold only for the busy beaver regime, which involves runtimes that are not physically realizable. While they provide insight into the fundamental algorithmic properties of objects and their effort-bounded compressibility, it is not immediately obvious how relevant these results are to real-world scenarios. To address this, we conduct empirical experiments that mirror our theoretical results. We choose three fundamentally different, yet still universal and practical, computational paradigms that represent a wide variety of possible universal computers: 2-Tag systems, elementary cellular automata with Rule 110, and the Brainfuck (BF) language.

In each of these systems, we run all programs up to a certain length with generous runtime limits. This allows us to construct real-world runtime vs. complexity profiles for all objects (i.e., output strings) that are computed by multiple programs with different runtimes. From these profiles, we can compute the relationship between steps since the last observed progress and the empirically achieved further compression progress (the equivalent to $\hat{k} - k_x$). We also compute the number of steps between the last observed progress, and the last overall progress (the runtime equivalent of $m_x - \hat{m}$). Using importance sampling, we re-weight objects based on their prior probabilities and plot the curves in Figures 10, 11, and 12. These are the empirical equivalents of the theoretical expectations shown in Figure 4.

Recent compression progress implies further compression progress in the future, especially under the assumption of the Algorithmic Prior. The most significant difference from the theoretical results is with respect to the Speed Prior: in the heavily constrained empirical setting, the logarithmic bias against long runtimes is less pronounced. This is especially the case for Rule 110 cellular automata, for which—due to their construction—no programs with very short runtimes exist, because programs typically require a "warm-up" period to generate complex glider interaction. The exact experimental setups for the three systems are detailed below[3].

**2-Tag**    (Figure 10) In this system, we run 2-Tag systems (Post, 1943) with the alphabet $\{a, b, c, H\}$, where $H$ is the halting symbol. These kinds of systems have been shown to be Turing complete (Wang, 1963; Cocke & Minsky, 1964). We enumerate all possible production rules up to a combined length of 13, with $H$ appearing only at the end of a single rule. The starting word is always '$aaa$'. This leads to approximately 120 million programs which we run for up to 100k steps. If no output is produced within this limit, we consider the program non-halting.

---

[3]Detailed interactive visualization of the results in this section can be found at https://inductive-interestingness-2026.github.io/future-compression-progress/#experiments

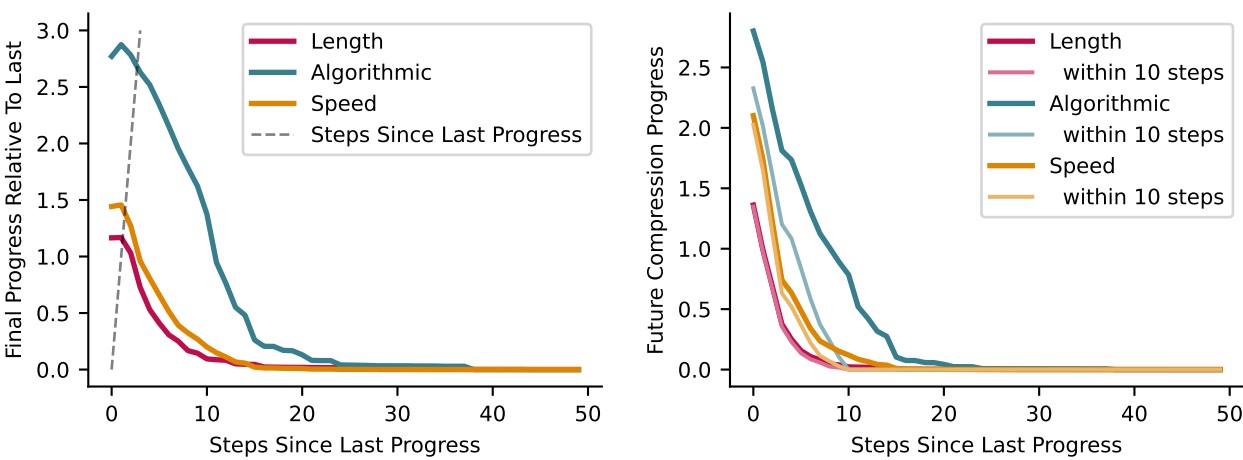

*Figure 10.* 2-Tag Systems. The average step of the final compression progress (left), and the average remaining compression progress (right), relative to the most recent progress. Plotted as a function of the steps since this most recent progress. As the number of these steps (the stagnation length) increases, the average remaining compression progress vanishes. Curves shown for outputs weighted with the Length, Algorithmic and Speed Prior. The right plot uses lighter shades to indicate near-term future progress occurring shortly after the observation cutoff. These results mirror the theoretical findings from Section 4.

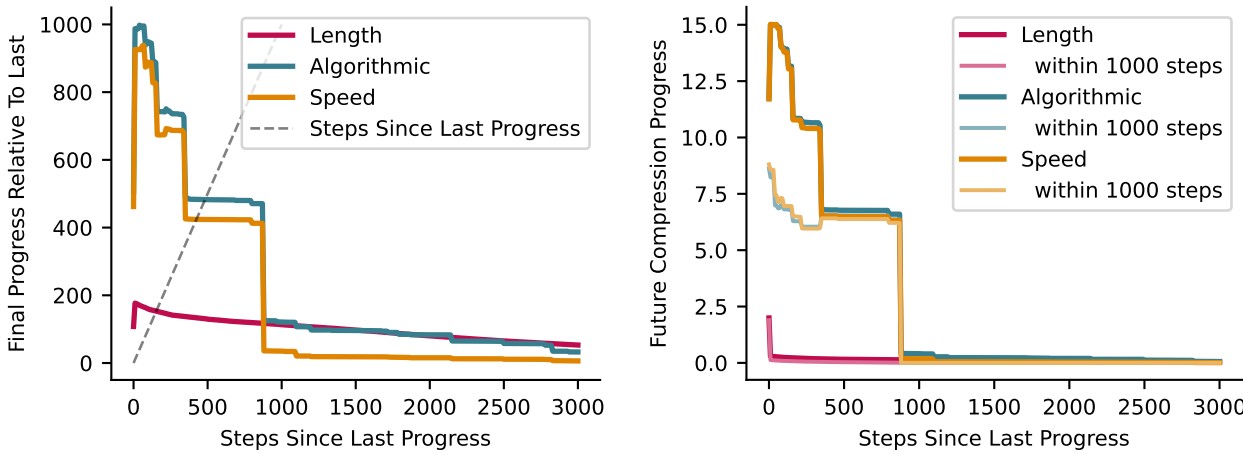

*Figure 11.* Rule 110. Analogous to Figure 10. Due to there not being any programs with very short runtimes, the difference between Algorithmic Prior and Speed Prior is less pronounced.

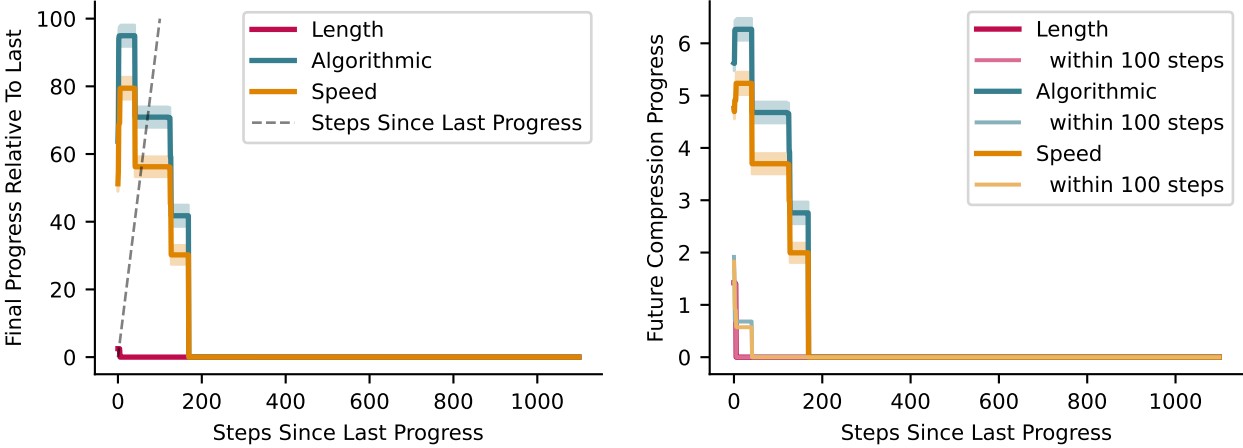

*Figure 12.* BF. Analogous to Figures 10 and 11. In all three figures, results are plotted with the standard error. It is noticeable only here.

**Rule 110**  (Figure 11) Rule 110 is a Turing-complete elementary cellular automaton (Wolfram, 1983; Cook et al., 2004). We use a state of size 512 with a cyclic boundary condition[4]. The programs are defined as the leftmost bits of the tape, up to the last value of 1. We enumerate all programs up to length 25, resulting in approximately 34 million simulations. A program halts as soon as any particular state appears for a second time (which means that there is a cycle). This state is then the output of the program. If no state repeats within 100k steps, we consider the program non-halting.

**BF**  (Figure 12) BF (Müller, 1993) is a highly compact Turing-complete programming language. Since we are not providing any external inputs, we omit the *read* instruction ',''. We enumerate all programs up to length 11 consisting of the remaining 7 instructions, leading to approximately 2.3 billion different programs, which we run for up to 100k steps.

---

[4]We do this purely for practical reasons and are aware that this technically reduces the automaton to a finite state machine.

