# OpenReview forum: "Position: Interestingness is an Inductive Heuristic for Future Compression Progress"
_ICML.cc/2026/Position_Paper_Track — ICML 2026 Position Paper Track regular_

### Official Review · Reviewer_fEks · 2026-03-06

**Significance:** 4
**Argument Clarity:** 2
**Rating:** 5
**Confidence:** 2

**Questions:**

I do not have specific questions to which the answer would change my opinion (see weaknesses), and I understand that the paper primarily provides a new framework, poses new questions, and inspires new research. However, I am curious whether there already exist algorithms, methods, or concrete examples that partially implement these ideas or would be a suitable starting point for further research. A concrete example would help to embed this paper in some practical context.

**Alternative Views Section:**

Yes

**Compliance With Llm Reviewing Policy A Conservative:**

Affirmed.

**Discussion Potential:**

3

**Final Justification:**

After reading the rebuttal and other reviews, I believe this paper, although quite dense in technical details, discusses a novel and interesting perspective. The rebuttal addressed my concerns, and I think that, with the additional illustrations, the paper would provide an interesting basis for discussion at ICML. Thus, I'll keep my positive score.

**Paper Summary:**

The paper proposes to quantify and leverage interestingness as a predictive heuristic for progress. For this they leverage information theoretic tools to build a framework that studies progress using complexity-runtime profiles showing that future progress can be estimated based on past observation and under specific assumptions. The paper furthermore discusses what is needed to satisfy these assumptions towards open-ended learning.

**Position:**

Yes

**Position In Title:**

Yes

**Related Work:**

4

**Strengths And Weaknesses:**

**Strengths**

1) The paper provides a theoretically grounded perspective on studying and assessing progress in AI. The paper presents several ideas that are engaging to a broad audience.
2) The alternative view sections are very useful to understand the position and also connect it with ideas from other research fields.

**Weaknesses**

The runtime and complexity profiles form the basis for the paper's main argument. While the introduction and conclusion are easy to follow and very well written, Sections 3 and 4 are very dense and not accessible to readers with a different background. I believe this section would benefit from more meta-structure and clearer explanations of notation to make the insight accessible to an even broader audience (for whom this position paper is highly relevant). Some examples:

  * (p4; 188-208r) U, p, and o are introduced, but never used. What is c?
  * (p4/5) The connection between the intro of Section 3 and 3.1 is not immediately clear (it becomes clear, but only at the end of Section 3)
  * (p 7) The paper provides an intuitive explanation of the length prior and the algorithmic prior, but not for the speed prior.

Minor comments (not taken into account for rating):

  * (p4 168l) "What we usually do when we have ..." --> something is missing/wrong in this sentence
  * (p7 350l) "The speed prior the is defined as" --> remove "the"

**Support:**

3

---

> ### Author Rebuttal · Authors · 2026-03-31
>
> Thank you for your assessment and helpful comments. We agree that Sections 3 and 4 are dense for readers without a background in Algorithmic Information Theory. We have since overhauled these sections to improve the structure and provide clearer definitions. To answer your specific notation question: $c$ denotes the Kolmogorov complexity (description length), so the y-coordinate of the points constituting the $D_x$ profile. We also appreciate the careful proofreading and have corrected the typos on pages 4 and 7.
>
> Since the submission, we have worked on making these concepts more accessible to a broad machine-learning audience: We generated interactive diagrams to help readers visualize how complexity and runtime profiles interact, and how the observed profile shape influences the expectation of further progress ([https://inductive-interestingness-2026.github.io/future-compression-progress/#theory](https://inductive-interestingness-2026.github.io/future-compression-progress/#theory)). Furthermore, we added experiments demonstrating that our results hold not only in the theoretical limit for large runtimes, but also empirically for short runtimes across three different universal compute paradigms ([https://inductive-interestingness-2026.github.io/future-compression-progress/#experiments](https://inductive-interestingness-2026.github.io/future-compression-progress/#experiments)).
>
> ## Speed Prior Intuition
> We have added an intuitive explanation for the Speed Prior: One problem with the Algorithmic Prior is that, once a program is chosen, we cannot know if it will ever halt. The Speed Prior resolves this by having the "monkey" first type a maximum runtime (in prefix-free format) and then the program. The program is run only for that specified time. If it hasn't halted, it returns an empty string. The Speed Prior is simply the probability that this time-bounded setup produces a specific string.
>
> ## Existing Methods and Practical Context
> There are several works that partially implement the ideas of our framework, starting from early work on information gain [1], which introduces an interestingness criterion as intrinsic reward, optimized for using reinforcement learning techniques.
>
> A good recent fit is **MAGELLAN** [2]: This work, similarly to several other works from the modern Learning Progress literature (for an overview, please see [3]), matches many of the ideas discussed in our paper. It implicitly relies on the assumption of inductive interestingness, which we formalize: Future learning progress is predicted by current learning progress. However, we highlight a key nuance: In their method, future learning progress is based on the *magnitude* of past learning progress. Our results suggest that the *recency* of the last non-trivial progress is more important than its magnitude. Tracking this recency of progress is what we mean by modeling the "compression frontier". How to efficiently do that is one of the challenges we pose to the community.
>
> Another recent work is [4], which measures the in-context information gain in LLMs and treats it as an interestingness criterion, using it to measure task complexity and to enhance performance in creative generation tasks.
>
> [1] Storck, J. et al. "Reinforcement Driven Information Acquisition In Non-Deterministic Environments", ICANN, 1995.
>
> [2] Gaven, L. et al. "MAGELLAN: Metacognitive predictions of learning progress guide autotelic LLM agents in large goal spaces." arXiv, 2025.
>
> [3] Colas, C. et al. "Autotelic Agents with Intrinsically Motivated Goal-Conditioned Reinforcement Learning: A Short Survey", JAIR, 2022.
>
> [4] Herrmann, V., Alcaide, E. et al. "Multiple Token Divergence: A Measure of In-Context Computation Density", ICLR 2026.
>
> ---
>
> We hope that this helps clarify our points and are very happy to discuss any remaining questions or concerns!

---

> > ### Author Rebuttal · Reviewer_fEks · 2026-04-02
> >
> > Thanks a lot for your response and additional information. I keep my score and confidence.
> >
> > While the visualizations seem helpful for future readers, I did not consider them. See rebuttal guidelines: `[...] links may only be used for figures (including tables) and captions that describe the figure (no additional text)`.

---

### Official Review · Reviewer_dTXR · 2026-03-15

**Significance:** 4
**Argument Clarity:** 2
**Rating:** 4
**Confidence:** 4

**Questions:**

Please see weaknesses. Thank you.

**Alternative Views Section:**

Yes

**Compliance With Llm Reviewing Policy A Conservative:**

Affirmed.

**Discussion Potential:**

4

**Final Justification:**

The rebuttal addressed most of my concerns.

**Paper Summary:**

The paper advocates for the engineering of scale-free synthetic environments as a pathway to truly autonomous open-ended systems.

**Position:**

Yes

**Position In Title:**

Yes

**Related Work:**

2

**Strengths And Weaknesses:**

Overall, I like the paper. It has interesting positions that are certainly worthwhile to be discussed. However, there is room for improvement in the writing and explanation, and how recent works fit into the proposed framework.

Strengths:
- I think this is a timely paper with lots of interesting discussion points.
- I appreciate the theoretical analysis and alternative views of interestingness.

Weaknesses:
- Writing can be improved. I list down concrete areas where I think writing can be improved below.
- In section 2, while I can infer the transition from one paragraph to another, the transition is not really explained well.
  - For example, from the “count-based, space coverage, and maximum entropy” paragraph to “prediction error, adversarial criterion” paragraph, the transition sentences of “If there are more possible objects than can ever be encountered, this measure of interestingness completely depends on some form of coarse-graining or smoothing. In other words, it depends on a model.” it talks about the limitation of count-based methods, and hence requiring some form of model to embed or interpret what the space of objects look like. Hence, the next paragraph talks about current methods of models doing that. This transition can be inferred but it is not well explained. In these sentences, what “coarse-graining or smoothing” means is not clear. They are opposing adjectives and are hence confusing when used together. The paper most likely mean that in some cases, we want to discretize/ down-sample/ coarse-grain the space, and in some cases we want to smooth the space. Why and when we should do each is unclear.
  - In the paragraph on “prediction error, adversarial criterion”, “this notion moves beyond …”. “this notion” is not yet specified.
  - Again, when transitioning to the paragraph “information gain, bayesian surprise”, I can infer that “common approaches to tackle this problem…” probably means that the next paragraph will talk about the common approaches, but this transition is not clear. Seeing the first sentence of the “information gain” paragraph does not make it clear that we are now reading about the common approaches to tackle the problem.
  - When transitioning to “compression progress”, it says “It can be argued that the success of the NLL as a training objective for models is due to the strong connection between learning and compression”. It is not well-explained why successfully optimizing for NLL is due to strong connection between learning and compression. An example or more explanation will help me.
- The reference of “subjects” and “objects” give some form of intuition, but how does this reference fit into specific examples? For example, an AI system learning to code, what are the objects here? the skill of coding? Coding could have many different sub-skills and levels, when is it an object? It is not clear how this example fits into the framework. Providing more examples would help.
- Equations can be explained better
  - In line 212, the usage of c is confusing. is c a number or a function? In the definition of D_x, c seems like a number, but right afterwards, we say that c(r) is the boundary of the profile.
  - In line 217, what is capital R?
  - (Section 3.3) Where do n_P and k_P correspond to in figure 2?
  - What is the intuition beyond the complexity profile? Having examples to bring the reader through the equations will help.
  - Making clear in the figures which parts are synthetic and which parts are theoretical guarantees will help.
  - In line 227, what does the flipped P sign mean?
- In lines 387 - 394, the paper talks about structured domains and structuring the “world”. This point seems related to the previous “coarse-grained/ smoothing” point, which was not well-explained either. It is unclear when a domain is considered structured (line 391).
- In line 423, when talking about synthetic data, “if the generated artifacts lack the depth found in natural objects”. When does this happen? If we are working in a turing-complete space (e.g., code, RNN) or a space rich enough to simulate the real world (e.g., genie), how does generating synthetic data in such a space apply to this point? Or if a synthetically generated artifact (e.g., AI generated songs or pictures) can make humans think that a human created it, does it still lack the depth found in natural objects? How does this depth relate to “post-hoc” interestingness?
- Where does the concept of a system’s memory fit into this framework?
- I appreciate the call for architectures that model their own compression frontier. There is some recent work that does model this (although not all in one neural network). For example, [1] predicts the learning progress of an agent, [2] updates what is interesting or not via an archive. Evolutionary algorithms that continuously expand [3] implicitly model the compression frontier by choosing based on some heuristics. Are these works already considered as having introspective assessment?
- I appreciate the call for engineering scale-free environments. In line 429, “the fundamental principles for generating scale-free artifacts that exhibit emergence across multiple orders of magnitude”. When is something considered a new magnitude? Who or what should be an observer of these magnitudes?
- The biggest question I have for this paper’s advocate is about scale-free synthetic environment. What is missing in current environment generation algorithms? If we take a turing-complete space (e.g., code, RNN, game of life), what are the limitations in current methods to reach this scale-free environment that the paper is advocating for?
- (nitpick that can probably be fixed) Line 44 says “To achieve true open-endedness, this cycle must be fully autonomous.” The co-evolution of technology and humans is open-ended, but that process involves humans, and hence not fully automated. When the paper says the cycle must be fully autonomous, the paper seems to have certain observers in mind or that the definition of open-endedness is referring to open-ended AI systems.

[1] Gaven, Loris, et al. "MAGELLAN: Metacognitive predictions of learning progress guide autotelic LLM agents in large goal spaces." arXiv preprint arXiv:2502.07709 (2025).

[2] Dharna, Aaron, Cong Lu, and Jeff Clune. "Foundation model self-play: Open-ended strategy innovation via foundation models." arXiv preprint arXiv:2507.06466 (2025).

[3] Tjanaka, Bryon, et al. "Discount Model Search for Quality Diversity Optimization in High-Dimensional Measure Spaces." arXiv preprint arXiv:2601.01082 (2026).

**Support:**

2

---

> ### Author Rebuttal · Authors · 2026-03-31
>
> Thank you for the very detailed and insightful comments. While we do not have space to acknowledge all suggestions here, they are immensely valuable for improving our paper.
>
> ## On Subjects and Objects
> In many cases, the distinction between subject and objects is fairly straightforward: The subject is the model and its learning algorithm, the objects are the training data. In reinforcement learning setting, the subject is the agent, and objects are—depending on the setup—trajectories, skills, tasks, environments or goals. So for a coding agent, the objects might either be the source code itself, or specific coding skills. For our theoretical setting, anything that can be expressed as a binary string could be an objects.
>
> ## Theoretical & Empirical Results
> We have addressed the issues in the notation and created an online visualization tool that should make the theoretical definitions, concepts and connections easier to grasp (https://inductive-interestingness-2026.github.io/future-compression-progress/). Furthermore, since our theoretical results give guarantees only for unrealistically large runtimes, we ran empirical experiments showing that our results hold for small runtimes in three very different Turing complete computational domains. These should further clarify the meaning of the curves. In a standard machine learning setup where neural network is trained using an NLL objective, the Complexity vs. Runtime profile (its lower boundary $c(r)$, to be precise) represents approximately the loss curve over training steps.
>
> Regarding the question of memory: The log-size vs. complexity profiles are based on two-part descriptions. What is plotted is the size of the model (i.e., the description length of its functionality and its memory) on the x-axis, and the complexity of the data, given the model, on the y-axis. Therefore, the profiles automatically take a model's memory into account.
>
> ## Comparison to Recent Work
>
> **MAGELLAN** is close in spirit to what were are proposing. Please see our reply to reviewer fEks for more details on that setting.
>
> **FMSP & DMS**: These fall under our "Alternative View" . While quality-diversity and self-play are powerful, they often lack the introspective assessment we advocate. It can be argued, however, that implicitly the diversity of the population/the archive is related to interestingness: Adding a policy to the archive that increases performance or adds a capability clearly is a form of learning progress. Using the inductive argument, we are now in a better position to make further progress. However, this interpretation arguable stretches the concepts we proposed to their limits.
>
> ## Scale-free Emergence
> The word magnitude in scale-free emergence is perhaps unnecessary, since again mean simply the complexity in bits of the model in a log-size vs. complexity profile (as detailed in [1]). Perfectly describing every little detail requires a lot of information, any emergent higher level description requires fewer bits. Using the $P \leftrightarrow D$ correspondence, the "magnitude" can also be measured in runtime steps.
>
> There is work on synthetically data using e.g. fractals [2], RNNs [3] or neural cellular automata [4]. Even in a Turing-complete domain, almost all generated data is "shallow", in the sense that not much can be learned from it. Hence, training on synthetic data works if a human expert carefully chooses the exact generation setting, so that it contains many levels of emergence. To move toward artificial open-ended system, we need to automate this process. To the best of our knowledge, there are no established criteria for scale-free emergence, or effective ways to detect and construct it. More research is needed understand whether it is possible to formulate exact conditions under which processes yield many levels of emergence. We point to the fields of Complex Systems, AIT, and perhaps Artificial Life (see e.g. [5]) for potential insights regarding that question.
>
> A note on AI generated songs and pictures: while they might superficially fool a human, we are not familiar with any fully autonomously AI generated artifacts that capture and deeply engage humans for extended periods of time. Some symphonies or paintings lead experts to new insights about them, even after hundreds of hours of engagement. Those are prime examples of scale-free emergent artifacts we do not know how to reliably produce.
>
> [1] Bédard C. A. & Bergeron, G., "An Algorithmic Approach to Emergence", Entropy, 2022
>
> [2] Kataoka et al., "Pre-training without Natural Images", ACCV 2020
>
> [3] Bloem, P., "Universal pre-training by iterated random computation", arXiv arXiv:2506.20057, 2025
>
> [4] Lee, D. et al., "Training Language Models via Neural Cellular Automata", arXiv arXiv:2603.10055, 2026
>
> [5] Bedau, M. A. et al. "A classification of longterm evolutionary dynamics", ALife VI, 1998
>
> ---
>
> We are very happy to engage in further discussion and hope this clarification addresses your concerns.

---

> > ### Author Rebuttal · Reviewer_dTXR · 2026-04-04
> >
> > I thank the authors for the detailed reply. I have some follow up questions, and some of my concerns were not addressed.
> >
> > "anything that can be expressed as a binary string could be an object"; In this sense, all computable programs can be expressed as binary strings, when does a binary string become a subject? E.g. a neural network model can be expressed as a binary string.
> > "So for a coding agent, the objects might either be the source code itself, or specific coding skills"; what about a self-improving coding agent (eg, the Darwin Godel Machine [1])?
> >
> > The rebuttal does not clarify the distinction between "coarse-graining" and "smoothing".
> >
> > The rebuttal acknowledges that there are no established criteria for scale-free emergence. Given this, what concrete research directions or measurable proxies do the authors suggest for progressing toward this goal?
> >
> > [1] Zhang, J., Hu, S., Lu, C., Lange, R., & Clune, J. (2025). Darwin godel machine: Open-ended evolution of self-improving agents. arXiv preprint arXiv:2505.22954.

---

### Official Review · Reviewer_FCp8 · 2026-03-16

**Significance:** 3
**Argument Clarity:** 3
**Rating:** 4
**Confidence:** 2

**Questions:**

Can the authors give a precise definition of "open-endedness"? I was expecting a discussion around its meaning in the paper, but it is always implicit. Do the authors intend a form of continual learning or some generalization of learning that treats models which progressively learn to solve new tasks?

As far as I understood, the introduction of BB is meant to detach the level of interestingness from a specific model implementation. But, how does this meet the practice? Would it not be the case, as in active learning, that we provide models with data where they model some kind of uncertainty? Can you also expand on the relation to Active Learning?

Because of all quantities depend on a notion of AIT complexity, computing them might not be trivial. How can we overcome these issues and actually the notion introduced to guiding a search of "interesting artifacts"?
Can the authors give a concrete example in which setting this search can be helpful

**Alternative Views Section:**

Yes

**Compliance With Llm Reviewing Policy A Conservative:**

Affirmed.

**Discussion Potential:**

2

**Final Justification:**

The paper presents an interesting idea which can be implemented to guide interestingness.

Overall, my score is positive, but some details are still too far-reaching for a position paper.

**Paper Summary:**

This position argues that a measure of "interestingness", grounded on Algorithmic Information Theory (AIT), is a necessary notion to deal with open-ended learning. The authors propose a formal argument based on AIT that future progress (measured as increasing compression) can be guided by detecting instances which increase the "surprise" of the model in information-theory sense and can elicit data to learn that contribute to a better structure (i.e. simpler). This view is contrasted to settings of multi-agent systems, where open-endedness emerges to some degree, and to settings that do not require AIT measures, e.g. leveraging LLMs as judges.

**Position:**

Yes

**Position In Title:**

Yes

**Related Work:**

3

**Strengths And Weaknesses:**

**DISCLAIMER** I am not an expert in AIT, so my review is an educated judgment based on my knowledge and on the position itself.

### STRENGTH

The position appears to me solid for a formal perspective (tracing all relevant material from more recent papers in AIT) and grounded on Kolmogorov Complexity, which is examined for AGI and to describe emergent behavior beyond AI (see Physics).  The appealing argument is that, following AIT, the correspondence (almost affine transformation) to BB function hold the potential to measure the "intrinsic algorithmic nature of the objects, rather than to ... a reference". This might be very interesting to relate to emergence of properties in language models, by properly identifying which data offer (even retrospectively) a better compression, or generalization.
This principle can likely guide the design of training setups for machine learning models and while, it seems still far to reach, can design models that predict their own learning progress. Second, this can guide understand which "scale-free artifacts" can guide emergence of models' capabilities.

All in all, this can guide some discussion on how the AI community should think about these artifacts when equipped with ML models, and so potentially shed light in what properties are expected out of them after completing training.


### WEAKNESSES

Premising that I could not fully understand some details in Section 3, the technical presentation obfuscates a bit the main message. In particular, the notation is sometimes implicit in the text, e.g. 3.3 starts with P, but it is not clear the connection to Px. The same section connects to existing works, but determining the meaning of all the quantities introduced is overwhelming. This, in turn, complicates the interpretation of the results in 4.1.

Despite these lacks, the position feels somehow very abstract and not easy to implement. The notions of AGI and open-endedness are precisely vague in the paper and so it is hard to trace a specific challenge in current literature which the authors have in mind. Similarly, the call to action is more at a high-level and does not propose precise directions, but an overall indication what is the correct thing to learn.
Because the position lacks the links to current methodologies in the community, I cannot easily say to which part of the community the position is relevant and how the discussion on "interestingness" can guide towards the authors goal (see "we can begin to build systems that do not merely solve the tasks we give them ...").

**Support:**

2

---

> ### Author Rebuttal · Authors · 2026-03-31
>
> Thank you for your thoughtful review. We appreciate the perspective of a reader outside the AIT community, as the goal of this position paper is to bridge these theoretical foundations with broader AI research.
>
> ## Mathematical Notation and Results
>
> We agree that the clarity of our mathematical notation could be improved, and we have overhauled Section 3 to ensure all quantities are explicitly defined. Addressing the concrete example, $P$ is a generic valid profile, wheres as $P_x$ is the profile of a specific string $x$ ($x$ is sampled from one of the prior distributions).
>
> To make these AIT concepts more intuitive, we created interactive diagrams that visualize these profiles, their relation, and the expected continuations (https://inductive-interestingness-2026.github.io/future-compression-progress/#theory). Furthermore, we now provide empirical corroboration of our results (see below).
>
> ## Definitions of AGI and Open-Endedness
>
> While AGI is notoriously difficult to define, we think it is reasonable to assert it involving major scientific and creative breakthroughs without human assistance. While perhaps there is also no universally agreed upon definition of Open-Endedness, we operationally accept the definition from the 2024 ICML position paper [1]: The ability of a system to continuously come up with artifacts that *novel and learnable* to an observer. Other works have discussed the importance of open-endedness for achieving AGI [2,3,4]. The idea is that, in order to continuously generate novel valuable insights, we need *divergence* instead of *convergence*. An that an "interestingness criterion" might be the necessary driver for this divergent learning process.
>
> We believe our paper is of particular interest to the established sub-fields of *open-endedness*, *recursive self-improvement*, and *intrinsic motivation & artificial curiosity*—or more broadly, to anyone who is trying to build systems that resemble the self-sustaining learning cycle discussed in the Introduction.
>
> [1] Hughes, E., Dennis, M., et al. "Position: Open-Endedness is Essential for Artificial Superhuman Intelligence", ICLR, 2024.
>
> [2] Stanley, K.O., Lehman, J. "Why Greatness Cannot Be Planned: The Myth of the Objective", Springer, 2015.
>
> [3] https://www.oreilly.com/radar/open-endedness-the-last-grand-challenge-youve-never-heard-of/
>
> [4] Zhang, J., et al. "Omni: Open-endedness via Models of Human Notions of Interestingness", ArXiv, abs/2306.01711, 2023.
>
> ## The Busy Beaver Function and Practicality
>
> The Busy Beaver function indeed allows us to talk about very general algorithmic trends, without worrying about implementation details. More concretely, it abstracts away the potentially exponential or even greater differences in runtime between various universal computers. This comes at the cost of our theoretical results holding only for unrealistically large runtimes. We recognize that this is a limitation. Because of that, we now provide additional empirical experiments: https://inductive-interestingness-2026.github.io/future-compression-progress/#experiments. We show that the property of inductive interestingness (or more concretely, that the recency of the last compression progress exponentially affects the expected further compression progress) holds empirically also for small runtimes in three widely different universal computation paradigms. Now we have shown inductive interestingness empirically on the very small scale, using exhaustive program search, and theoretically in the large scale limit. We believe it is reasonable to assume that it holds also in the middle ground, where most current machine learning methods are located. This is our call to the community: test the idea of inductive interestingness—track the recency of progress or insight, use it to find identify promising artifacts—and try to use it to build systems that continually learn and discover new insights.
>
> ## Relation to Active Learning
>
> Under default assumptions, active learning asks: Which pieces of data from a fixed unlabeled dataset should be labeled to maximize a model's performance on a pre-defined task? Our question is: Out of all possible computable objects, which hold promise of providing novel insights when engaging further with it? These questions are clearly related, but not identical. Many of the criteria we discuss in Section 2 are also used in Active Learning. Basing interesting purely on uncertainty can suffer from the same "Noisy TV" problem as the Adversarial Criterion described in the paper. Since most approaches of active learning do not take any history of progress into account, they are not close to our notion of inductive interestingness.
>
> ---
>
> We believe these clarifications and the addition of empirical evidence address your main concerns regarding the abstract nature of the work. We would be very happy to discuss further and hope you might consider raising your score in light of these improvements.

---

> > ### Author Rebuttal · Reviewer_FCp8 · 2026-04-03
> >
> > I thank the authors for the clarifications and explaining the connection to Active Learning.
> > I give merit to the inclusion of more concrete evidence via experimental prototypes. I will raise by one point my score.

---

### Official Review · Reviewer_RNQM · 2026-03-22

**Significance:** 2
**Argument Clarity:** 2
**Rating:** 2
**Confidence:** 4

**Questions:**

Please see the Weaknesses section.

One additional question for calibrating on scope and falsifiability: among these systems, which do not constitute a faithful implementation of your notion of interestingness, and why?
- LLM with a linear head predicting loss after N more gradient steps
- Active learning with uncertainty sampling
- AlphaZero playing against itself https://arxiv.org/abs/1712.01815
- Altering data mixtures by fitting scaling laws https://arxiv.org/abs/2403.16952
- Using a world model to maximize info gain over imagined trajectories https://arxiv.org/abs/2005.05960

**Alternative Views Section:**

Yes

**Compliance With Llm Reviewing Policy A Conservative:**

Affirmed.

**Discussion Potential:**

2

**Paper Summary:**

This paper argues that open-ended learning (and AGI) requires a formalized notion of *interestingness* as a crucial ingredient. For a system to be truly autonomous, it must generate its own problems, and this paper argues for interestingness as a way to enable this. In very rough terms, interestingness seems to be a combination of novelty (different from what has been seen before) and learnability (contains structure that the model _can_ compress). They define interestingness by the progress of future compression. Their proposal is grounded in algorithmic information theory, specifically complexity-vs-runtime profiles. They also analyze the inductive properties of interestingness under three priors, showing that interestingness is sensitive to priors. In practical terms, they warn against leaning too heavily on synthetic environments or data, and note that the real world exhibits scale-free emergence, where interestingness is a useful heuristic.

**Position:**

Yes

**Position In Title:**

Yes

**Related Work:**

3

**Strengths And Weaknesses:**

## Strengths

- The taxonomy in Section 2 cleanly contrasts existing criteria (count-based, prediction error, information gain, compression progress) and argues that each is retrospective. To my best knowledge, the claim that interestingness should be a _prospective_ heuristic is novel.
- Section 4 shows that interestingness gives qualitatively different predictions under different priors. Sensitivity to assumptions on the prior is an explicit, quantified finding of this paper.
- Section 5's inversion (asking what properties a *distribution* must have for interestingness to be inductive) directly informs synthetic data and self-play work.

## Weaknesses

- My main concern is that the position is unfalsifiable. "Interestingness" is defined broadly enough (as any prospective heuristic for future compression progress) that virtually any curriculum selection or data filtering method could be recast as an instance of it. The Alternative View boxes seem to absorb competing frameworks by arguing they are "really" special cases of interestingness. No criterion is given for what would constitute evidence *against* the thesis. _What would a world look like where interestingness is *not* essential for open-ended learning?_
- Somewhat related to the point above, the paper has no empirical grounding. What existing methods are closest to implementing this notion of interestingness? In which settings might we see the inductive property hold in practice, even approximately?
- The main practical recommendation is vague: "To achieve autonomy, we must develop introspective models capable of predicting their own learning progress." There is no specific mechanism proposed, nor a set of concrete criteria for when a system satisfies this property.
- "Scale-free synthetic environments": no testable criterion for when an environment is scale-free in the relevant sense. I understand the analogy to, e.g., the Mandelbrot set as a clear example of a scale-free environment, but what measurement of current synthetic data pipelines or environments would indicate they are not scale-free?

**Support:**

2

---

> ### Author Rebuttal · Authors · 2026-03-31
>
> Thank you for your rigorous critique. We appreciate the opportunity to clarify the boundaries and falsifiability of our position.
> ## Falsifiability
> Our position has several claims which are falsifiable and open to discussion:
> - **The Necessity of Interestingness:** We argue that open-endedness and AGI are bottlenecked by the prospective identification of learnable data. This is falsified if "naive scaling"—training on static, human-curated data without any autonomous selection or exploration strategy—successfully reaches AGI. Similarly, social multi-agent interactions (see the Alternative View section) might be successful. Then our framing clearly did not get at the heart of the problem.
> - **Interestingness is a heuristic for future compression progress:** This is a definition, and as such not directly falsifiable. We believe this is a novel and useful definition, but we are open to counterarguments (such as, Interestingness is not always just about the future, and perhaps not about compression at all).
> - **The Inductivity of Interestingness:** Our core theoretical finding is that the *recency* of the last "drop" in complexity is a more robust predictor of future progress than the magnitude or frequency of past progress. While our AIT results are mathematically certain, their transferability to deep learning is falsifiable. If the complexity-runtime profiles ($D_x$​) of neural networks do not exhibit the same inductive properties as our Turing-complete benchmarks, our framework's utility is limited. We now include initial experiments (see below).
>
> Similarly to most position papers we are familiar with, the ultimate test of our position is whether subscribing to it (or to any of the sub-positions) leads to fruitful research.
>
> ## Empirical Grounding
> We have added empirical experiments (https://inductive-interestingness-2026.github.io/future-compression-progress/#experiments) using exhaustive program search in simple Turing-complete paradigms. These results corroborate our theoretical claims: in these settings, the recency of compression progress exponentially affects the expectation of further progress. While they are not in the domain of machine learning, they make our abstract theoretical results and claims concrete. This now opens a wide space of methods that make use of inductive interestingness in more sophisticated way than our exhaustive program search. Many already existing methods occupy this space: Most intrinsic motivation and recursive self-improvement schemes aim at generating future insight, or learning and compression progress. For a discussion of concrete examples, please see our reply to Reviewer fEks.
>
> ## Specificity of "Introspective Models"
> By "introspective," we mean models that explicitly probe their own "compression frontier" along specific *effort axes* (e.g., gradient steps, thinking tokens, or model capacity) . A concrete mechanism would be a "meta-model" that tracks the $D_x$​ profile of encountered tasks: if increasing compute $r$ has recently yielded a drop in description length $c$, the task is prioritized.
>
> ## Scale-Free Criteria
> As we state in the paper, we believe that understanding, measuring, and creating scale-free emergence is an open challenge. We argue for its importance, and point to the science of Complex Systems, AIT, and Artificial Life as potentially promising avenues.
>
> ## Questions: Comparison to existing methods
> - **LLM with linear head predicting loss:** This does not count if it only looks at *content*. Our notion suggests a *history-based* assessment of the model's own compression trajectory .
> - **Active Learning:** We would not count standard uncertainty sampling is not ''interestingness'', please also see the reply to Reviewer FCp8.
> - **AlphaZero:** This is a counter-example that helps define our scope. In a fixed, perfect-information domain with a predefined winning criterion, "interestingness" is less vital. We care about general intelligence in open domains where structure is not initially known.
> - **Data Mixtures via Scaling Laws:** This fits well. Scaling laws are fundamentally inductive mappings of loss to effort. A direct application would be dynamically biasing data mixtures toward domains that have shown the most *recent* learning progress.
> - **Plan2Explore:** This approximates information gain via disagreement. While viable, it is a *population-based heuristic; we argue that a single agent can infer the same signal *inductively* from its own past progress history.
>
> ---
> We hope these clarifications demonstrate that our framework is a specific, testable paradigm for autonomous discovery, and might lead you to reconsider your rating. You might also be interested in our new interactive diagrams on the relationship between the shapes of $P_x$ and $D_x$ profiles, and their connection to expected compression progress (https://inductive-interestingness-2026.github.io/future-compression-progress/#theory). We look forward to further discussion.

---

> > ### Author Rebuttal · Reviewer_RNQM · 2026-04-05
> >
> > Thank you for the detailed rebuttal and supplementary material. My concerns about falsifiability are addressed. The additional experiments also make the theoretical claims more concrete than in the original submission. The rebuttal also clarified the scope of what the paper views as instances of "interestingness".
> >
> > On the “scale-free” point, I appreciate the candor that this is an open challenge, but that also means one of the paper’s main practical warnings remains difficult to evaluate or act on.
> >
> > That said, my main concerns remain largely unresolved. In particular, I still do not find the central position operationally falsifiable in a useful sense. The proposed falsifiers, such as naive scaling reaching AGI, are coarse and distant, and the framework still appears broad enough to accommodate many competing approaches. Relatedly, I would argue that current post-training recipes, and perhaps many other reasonable approaches, already instantiate this principle by adaptively allocating resources toward data, environments, or feedback signals that work well with the current model. That makes it less clear what would distinguish this framework from a broad family of existing heuristics, as opposed to redescribing it at a higher level of abstraction.

---

### Decision · Program_Chairs · 2026-04-30

**Decision:**

Accept (regular)

**Comment:**

I think the paper takes an interesting view, and there is value in some of the arguments brought up. It provides a formal argument rooted in Kolmogorov complexity. The authors rebuttal provides detailed answers to most of the raised concerns, reflected in the reviewers willingness to raise their score.

That said, I do think there is a certain level of specificity or clarity somewhat missing from the paper. Specifically I think interestingness is widely defined, which makes difficult to not agree with the position. Therefore most of the value of the work rests in the careful formalism and  in the argument being made itself.